# Determinant Estimation under Memory Constraints and Neural Scaling Laws

Siavash Ameli [1 2]   Chris van der Heide [3]   Liam Hodgkinson [4]   Fred Roosta [5]   Michael W. Mahoney [1 2 6]

## Abstract

Calculating or accurately estimating log-determinants of large positive definite matrices is of fundamental importance in many machine learning tasks. While its cubic computational complexity can already be prohibitive, in modern applications, even storing the matrices themselves can pose a memory bottleneck. To address this, we derive a novel hierarchical algorithm based on block-wise computation of the LDL decomposition for large-scale log-determinant calculation in memory-constrained settings. In extreme cases where matrices are highly ill-conditioned, accurately computing the full matrix itself may be infeasible. This is particularly relevant when considering kernel matrices at scale, including the empirical Neural Tangent Kernel (NTK) of neural networks trained on large datasets. Under the assumption of neural scaling laws in the test error, we show that the ratio of pseudo-determinants satisfies a power-law relationship, allowing us to derive corresponding scaling laws. This enables accurate estimation of NTK log-determinants from a tiny fraction of the full dataset; in our experiments, this results in a $\sim$100,000$\times$ speedup with improved accuracy over competing approximations. Using these techniques, we successfully estimate log-determinants for dense matrices of extreme sizes, which were previously deemed intractable and inaccessible due to their enormous scale and computational demands.

[1]Department of Statistics, University of California, Berkeley CA, USA [2]International Computer Science Institute, Berkeley CA, USA [3]Dept. of Electrical and Electronic Engineering, University of Melbourne, Australia [4]School of Mathematics and Statistics, University of Melbourne, Australia [5]CIRES and School of Mathematics and Physics, University of Queensland, Australia [6]Lawrence Berkeley National Laboratory, Berkeley CA, USA. Correspondence to: Chris van der Heide <chris.vdh@gmail.com>.

*Proceedings of the $42^{nd}$ International Conference on Machine Learning*, Vancouver, Canada. PMLR 267, 2025. Copyright 2025 by the author(s).

## 1. Introduction

Many quantities of interest in machine learning require accurate estimation of the log-(pseudo-)determinant of large dense positive (semi-)definite matrices, often indexed by some number of datapoints. These quantities arise in a number of tasks, including training Gaussian processes (GPs) and other kernel-based methods (Wang et al., 2019), graphical models (Rue & Held, 2005), determinantal point processes (Kulesza et al., 2012), and model comparison and selection techniques (Hodgkinson et al., 2023b). Problems where the calculation cannot be avoided can often be reduced to computing a volume form, which is the case for tasks in statistical mechanics (Mézard & Montanari, 2009) or Bayesian computation (Gelman et al., 2013), or applications of the Karlin-McGregor theorem (Karlin & McGregor, 1959) including determinantal point processes. Similarly, when training GPs via empirical Bayes (Rasmussen & Williams, 2006), the log-determinant term is the most difficult to compute. In many applications, these matrices are not only dense but highly ill-conditioned. This renders methods that leverage sparsity inappropriate and makes accurate estimation of small eigenvalues essential, since errors in these small values are magnified at log-scale. Much of the research focus has concentrated on approximation techniques that ameliorate the cubic computational complexity of these log-determinant calculations. Methods leveraging stochastic expansions that rely upon matrix-vector multiplications—such as Lanczos-based methods (Ubaru et al., 2017)—have been particularly successful, yielding approximate methods that scale linearly (Dong et al., 2017), and methods leveraging Taylor's expansions (Fitzsimons et al., 2017) have also been proposed. However, accuracy can suffer when they encounter common ill-conditioning pathologies. Crucially, time complexity isn't the only bottleneck at play: large-scale matrices found in modern applications often hit the memory wall before computational cost becomes an issue (Gholami et al., 2024).

Recently, the empirical Neural Tangent Kernel (NTK) has become a prominent theoretical and practical tool for studying the behavior of neural networks both during training and at inference (Novak et al., 2022; Hodgkinson et al., 2025). The Gram matrix associated with the NTK has been used in lazy training (Chizat et al., 2019) and shows promise as a tool to obtain uncertainty quantification esti-

mates (Immer et al., 2021; Wilson et al., 2025). Its log-determinant—highly sensitive to small eigenvalues—has also received recent attention, both as a quantity of interest in model selection techniques that rely upon marginal likelihood approximation (Immer et al., 2023; Hodgkinson et al., 2023b) and as a way to quantify the complexity of a learning problem (Vakili et al., 2021). Similarly, both the NTK log-determinant and closely related quantities have recently appeared in quantification of generalization error via PAC-Bayes bounds (Hodgkinson et al., 2023c; Kim et al., 2023). While several approximations have been proposed (Mohamadi et al., 2023), convergence has only been shown in spectral norm, so they cannot be expected to capture the overall behavior of the full NTK.

In practice, computing the log-determinant of the empirical NTK's Gram matrix is a formidable task: besides suffering from the pathologies that ill-conditioned matrices are subject to at scale, the NTK appears to have its own peculiarities that make the problem particularly challenging. Indeed, *the task is considered so impenetrable as to be universally avoided*, since the NTK corresponding to most relevant datasets cannot be stored in memory. For example, storing the NTK corresponding to the relatively small MNIST dataset requires 2.9 terabytes of memory. The equivalent object for ImageNet-1k requires 13.1 exabytes, an order of magnitude larger than CERN's current data storage capacity (Smith, 2023). While it is appealing to consider only a small subset of the training dataset when making necessary approximations, naïve estimates for the log-determinant can incur significant bias. Furthermore, we will find that leading approaches for dealing with log-determinants of empirical NTK Gram matrices using sketching and other Monte Carlo approximations tend to be highly inaccurate.

Scaling laws have played a prominent role in machine learning theory and practice, providing insight into the asymptotic behavior of generalization error (Li et al., 2023; Vakili et al., 2021), as well as guidance for compute-optimal resource allocation when deploying deep learning models at scale (Kaplan et al., 2020; Hoffmann et al., 2022). It turns out that for large kernel matrices, the ratio of successive determinants can be cast in terms of the error of an associated Gaussian process. This enables known scaling laws to be deployed in estimating the log-determinant, with surprising accuracy. However in order to measure this accuracy against a reliable baseline, a memory-constrained method for exact out-of-core log-determinant computation is also required, and is of independent interest.

**Contributions.** The central contributions of this work address the issues faced when computing log-determinants of large matrices. We are primarily interested in matrices that cannot fit into memory. To compute an accurate baseline, we derive MEMDET, a memory-constrained algo-

*Table 1.* Comparison of stochastic Lanczos quadrature (SLQ, with degree $l$, $s$ Monte Carlo samples, and full re-orthogonalization), MEMDET (Algorithm D.2), and FLODANCE (Algorithm 1) on a dense $500,000 \times 500,000$ NTK matrix for a ResNet50 model trained on CIFAR-10 with 50,000 datapoints. MEMDET computes the exact log-determinant and serves as the benchmark, with relative errors of other methods measured against it. Costs and wall time are based on an NVIDIA H100 GPU ($2/hour) and an 8-core 3.6GHz CPU ($0.2/hour) using Amazon pricing and include NTK formation from a pre-trained network.

| Method | | | Rel. | Est. | Wall |
| Name | Settings | TFLOPs | Error | Cost | Time |
| --- | --- | --- | --- | --- | --- |
| SLQ | $l = 100, s = 104$ | 5203 | 55% | $83 | 1.8 days |
| MEMDET | LDL, $n_b = 32$ | 41,667 | **0%** | $601 | 13.8 days |
| FLODANCE | $n_s = 500,\ q = 0$ | **0.04** | 4% | $0.04 | 1 min |
| FLODANCE | $n_s = 5000, q = 4$ | 41.7 | **0.02%** | $4 | 1.5 hr |

rithm for determinant computation. This facilitates exact calculation of log-determinants of NTK Gram matrices corresponding to neural networks with several million parameters. We then provide a novel approximation technique, FLODANCE, based on the scaling behavior of a wide class of kernel matrices, by appealing to *neural scaling laws*. In detail, our main contributions are:

- we derive a hierarchical memory-constrained algorithm for large-scale computation of log-determinants, which we name MEMDET;

- under mild assumptions, we derive scaling laws for the ratio of pseudo-determinants of kernel matrices containing different subsets of the same dataset, enabling both a corresponding law of large numbers and central limit theorem for normalized log-determinants;

- leveraging these scaling laws, we propose FLODANCE, a novel algorithm for accurate extrapolation of the log-determinant from a small fraction of the full dataset;

- we demonstrate the practical utility of our method by approximating the NTK corresponding to common deep learning models; and

- we provide a high-performance Python package `detkit`, which implements the presented algorithms and can be used to reproduce the results of this paper.

Crucially, we demonstrate that our approximation technique is able to obtain estimates with lower error than incurred by reducing the numerical precision in explicit computation. Our approximation techniques render an impractical task *virtually routine*, as shown in Table 1. To the best of our knowledge, this is the first time that the full empirical NTK corresponding to a dataset of this scale has been computed.

The remainder of this document is structured as follows. In Section 2, we discuss the computational issues faced when computing determinants at scale, and we derive our MEMDET algorithm based on block LU computation of the log-determinant. Section 3 contains the appropriate scaling laws for pseudo-determinants of interest, the corresponding LLN and CLT for their logarithms, and the FLODANCE algorithm for their approximation. Numerical experiments are presented in Section 4, and we conclude in Section 5.

A summary of the notation used throughout the paper is provided in Appendix A. An overview of related work in linear algebra is given in Appendix B. Computational challenges when dealing with NTK matrices are then discussed in Appendix C. Implementation and performance analysis of MEMDET are provided in Appendices D and E. Required background for neural scaling laws, proofs of our theoretical results, and further analysis of FLODANCE appear in Appendices F and G. Comparison of various log-determinant methods is given in Appendix H. Finally, an implementation guide for detkit appears as Appendix I.

## 2. Computing Determinants at Scale

The computation of determinants of large matrices, particularly those expected to be highly ill-conditioned, is widely considered to be a computationally "ugly" problem, which should be avoided wherever possible (Axler, 1995). However, this is not always an option, e.g., when the determinant represents a volume form, is required in a determinantal point process, or plays a role in training GPs via empirical Bayes. Although certain limiting behaviors under expectation are well characterized (Hodgkinson et al., 2023a), they provide only a coarse approximation in finite-sample settings. The quadratic memory cost and cubic computational complexity of naïve implementations make exact computation prohibitive, necessitating the use of lower-precision or randomized methods at scale. While approximate methods can be useful in their own right, exact computation remains essential, at the very least to establish meaningful baselines. In our experiments, due to the pathological spectral behavior of the matrices we consider, we will see that our proposed exact method is comparable in speed to the state-of-the-art Monte Carlo approximation techniques.

### 2.1. Low-Precision Arithmetic

Among the most common techniques for circumventing memory and computational bottlenecks when dealing with large-scale calculations in numerical linear algebra is to cast numerical values into a lower precision, usually 32-bit, 16-bit, or even 8-bit floating point values, instead of 64-bit (double precision) values that would otherwise be used.

Computations of the log-determinant in mixed precision do not generally incur significant error. However, in our setting, the matrix of interest is realized as the product $\mathbf{J}\mathbf{J}^\mathsf{T}$, where $\mathbf{J}$ is the Jacobian. The formation of the quadratic is a well-known source of approximation error, and should be avoided if possible: paraphrasing Higham (2022), if $\delta$ is the round-off used in choice of floating point arithmetic, then for $0 < \varepsilon < \sqrt{\delta}$ we can consider the simple case of $\mathbf{J} = \begin{bmatrix} 1 & \varepsilon \\ 1 & 0 \end{bmatrix}$, we have

$$\mathbf{J}\mathbf{J}^\mathsf{T} = \begin{bmatrix} 1 + \varepsilon^2 & 1 \\ 1 & 1 \end{bmatrix}, \quad \mathsf{fl}(\mathbf{J}\mathbf{J}^\mathsf{T}) = \begin{bmatrix} 1 & 1 \\ 1 & 1 \end{bmatrix},$$

where $\mathbf{J}\mathbf{J}^\mathsf{T}$ rounds to the singular $\mathsf{fl}(\mathbf{J}\mathbf{J}^\mathsf{T})$. In general, significant precision loss should occur if $\mathsf{cond}(\mathbf{J}\mathbf{J}^\mathsf{T}) > \delta^{-1}$, where $\delta \approx 10^{-3}$, $10^{-6}$, and $10^{-15}$ for 16-bit, 32-bit, and 64-bit precisions respectively. In the Gram matrices we consider, $\mathsf{cond}(\mathbf{J}\mathbf{J}^\mathsf{T}) > 10^{12}$. Due to the scale of the problems of interest here, the sheer size of the Jacobian (requiring at least hundreds of terabytes of space) makes directly operating on $\mathbf{J}$ impractical. This necessitates breaking conventional wisdom and explicitly forming the Gram matrix. We will see later the incurred cost to accuracy when working in low precision.

### 2.2. MEMDET: A Memory-Constrained Algorithm for Log-Determinant Computation

Conventional methods for computing the determinant of matrices include LU decomposition (for generic matrices), LDL decomposition (for symmetric matrices), and Cholesky decomposition (for symmetric positive-definite matrices). These methods also simultaneously provide the determinants of all leading principal submatrices $\mathbf{M}_{[:k,:k]}$ of an $m \times m$ matrix (for $k = 1, \ldots, m$), possibly after permuting $\mathbf{M}$ depending on the decomposition.

The LU decomposition has a computational complexity of approximately $\frac{2}{3}m^3$, while both LDL and Cholesky decompositions have a complexity of approximately $\frac{1}{3}m^3$. While computational complexity is a concern for large-scale determinant computation, memory limitation poses an even greater challenge. These methods require substantial memory allocation, either the size of the array if written in-place, or twice the size if the input array is preserved.

To address this, we present MEMDET, a memory-constrained algorithm for large-scale determinant computation. Below is a sketch of the algorithm, with details of its implementation, memory and computational aspects provided in Appendix D, and a description of the software implementing this algorithm provided in Appendix I. We focus on the algorithm for generic matrices using LU decomposition, though the LDL and Cholesky decompositions follow similarly.

Consider the $2 \times 2$ block LU decomposition of $\mathbf{M}$ (see for

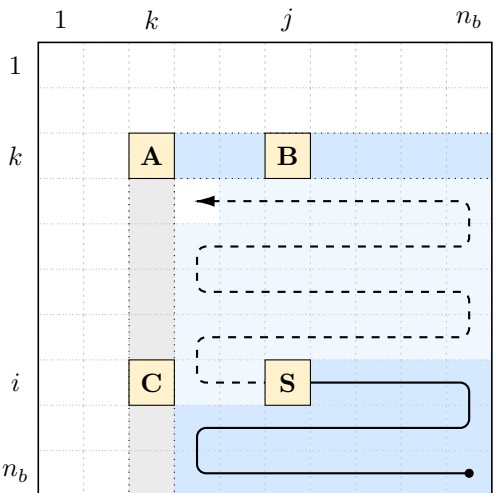 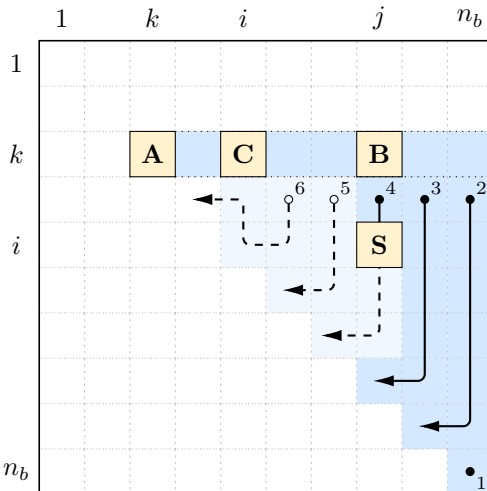

*Figure 1.* Schematic diagram of MEMDET illustrating the efficient block processing order for LU decomposition (Algorithm D.1, left panel) and LDL/Cholesky decompositions (Algorithm D.2, Algorithm D.3, right panel). The detailed ordering strategy is described in Appendix D.1.

instance, Dongarra et al. (1998b, Chapter 5.4)) with the first block, $\mathbf{M}_{11}$, of size $b \times b$, $b < m$, as

$$\mathbf{M} = \begin{bmatrix} \mathbf{M}_{11} & \mathbf{M}_{12} \\ \mathbf{M}_{21} & \mathbf{M}_{22} \end{bmatrix} = \begin{bmatrix} \mathbf{L}_{11} & \mathbf{0} \\ \mathbf{L}_{21} & \mathbf{I} \end{bmatrix} \begin{bmatrix} \mathbf{U}_{11} & \mathbf{U}_{12} \\ \mathbf{0} & \mathbf{S} \end{bmatrix}, \quad (1)$$

where $\mathbf{L}_{11}$ is lower triangular, $\mathbf{U}_{11}$ is upper-triangular, and $\mathbf{I}$ is the identity matrix. The blocks of the decomposition are obtained by computing $\mathbf{M}_{11} = \mathbf{L}_{11}\mathbf{U}_{11}$, solving lower-triangular system $\mathbf{U}_{12} = \mathbf{L}_{11}^{-1}\mathbf{M}_{12}$ and upper-triangular system $\mathbf{L}_{21} = \mathbf{M}_{21}\mathbf{U}_{11}^{-1}$, and forming the Schur complement $\mathbf{S} := \mathbf{M}_{22} - \mathbf{L}_{21}\mathbf{U}_{12}$. This procedure is then repeated on the $(m - b) \times (m - b)$ matrix $\mathbf{S}$, treated as the new $\mathbf{M}$, leading to a new $b \times b$ upper-triangular matrix $\mathbf{U}_{11}$ and a smaller Schur complement $\mathbf{S}$ at each iteration. This hierarchical procedure continues until the remaining $\mathbf{S}$ is of size $b$ or less, at which point its LU decomposition is computed. The log-determinant of the entire matrix is the sum of the log-determinants of all $\mathbf{U}_{11}$ blocks at each iteration.

To make this memory-efficient, the algorithm is modified to hold only small chunks of the matrix in memory, storing intermediate computations on disk. Suppose $\mathbf{M}$ consists of $n_b \times n_b$ blocks $\mathbf{M}_{ij}$ of size $b \times b$ where $i, j = 1, \dots, n_b$. We pre-allocate four $b \times b$ matrices $\mathbf{A}$, $\mathbf{B}$, $\mathbf{C}$, and $\mathbf{S}$ in memory. The $k$-th stage begins by loading $\mathbf{A} \leftarrow \mathbf{M}_{kk}$ from disk and performing an in-place LU decomposition $\mathbf{A} = \mathbf{L}\mathbf{U}$, with $\mathbf{L}$ and $\mathbf{U}$ stored in $\mathbf{A}$. We then compute the Schur complement for all inner blocks $\mathbf{M}_{ij}$, $i, j = k + 1, \dots, n_b$, by loading $\mathbf{B} \leftarrow \mathbf{M}_{kj}$ and $\mathbf{C} \leftarrow \mathbf{M}_{ik}$ from disk, solving $\mathbf{B} \leftarrow \mathbf{L}^{-1}\mathbf{B}$ and $\mathbf{C} \leftarrow \mathbf{C}\mathbf{U}^{-1}$ in-place, loading $\mathbf{S} \leftarrow \mathbf{M}_{ij}$, and computing $\mathbf{S} \leftarrow \mathbf{S} - \mathbf{C}\mathbf{B}$. The updated $\mathbf{S}$ is then stored back to disk $\mathbf{M}_{ij} \leftarrow \mathbf{S}$, either by overwriting a block of the original matrix, or avoids doing so by writing to a separate

scratchpad space. In the latter case, a cache table tracks whether a block $\mathbf{M}_{ij}$ should be loaded from the original matrix or from the scratchpad in future calls.

The computational cost of this procedure remains the same, independent of the number of blocks, $n_b$ (see Appendix E.1). However, increasing the number of blocks reduces memory usage at the expense of higher data transfer between disk and memory. Efficient implementation minimizes this by processing blocks in an order that reduces the loading of $\mathbf{B}$ and $\mathbf{C}$. The left panel of Figure 1 shows one such order, illustrating the procedure at the $k$-th iteration. Processing the blocks $\mathbf{M}_{ij}$ starts from the last row of the matrix and moves upward. During the horizontal and vertical traverses on the path shown in the figure, the memory blocks $\mathbf{C}$ and $\mathbf{B}$ can remain in memory, avoiding unnecessary reloading. Once the procedure reaches the block $(k + 1, k + 1)$, the memory block $\mathbf{S}$ can be directly read into $\mathbf{A}$ instead of being stored on disk as $\mathbf{M}_{k+1,k+1}$, initiating the block $\mathbf{A}$ for the next iteration. Consequently, only the blocks shown in blue need to be stored in the scratchpad space, while those in dark blue are already stored in the current state of the algorithm. This algorithm can be further modified to eliminate the need for $\mathbf{A}$ and use the memory space of $\mathbf{S}$ instead, but this would require storing the additional gray blocks on the scratchpad space in addition to the blue blocks. A pseudo code and further efficient implementation details of the presented method can be found as Algorithm D.1.

The algorithms for LDL and Cholesky decompositions follow a similar procedure with necessary adjustments for symmetric and symmetric positive-definite matrices, as detailed in Algorithm D.2 and Algorithm D.3. These modifications include processing only the lower (or upper) triangular part

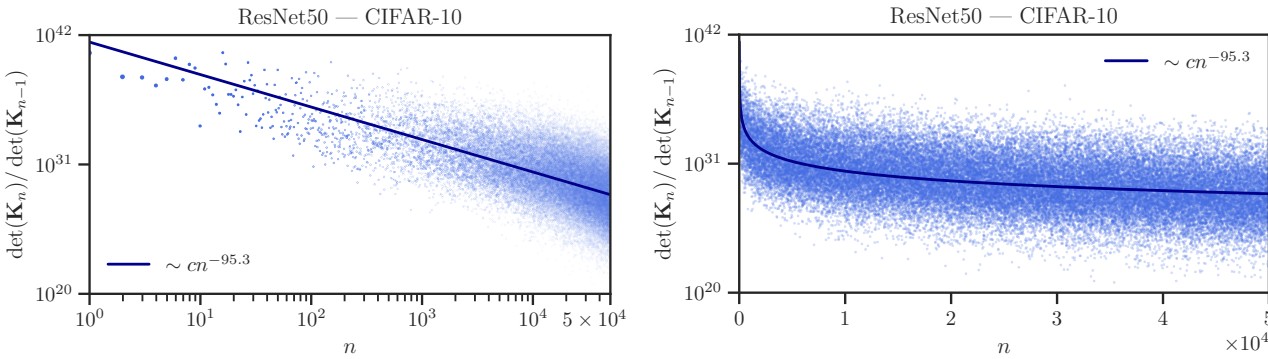

*Figure 2.* Demonstration of a scaling law for the ratios of successive determinants $\det(\mathbf{K}_n)/\det(\mathbf{K}_{n-1})$ (Assumption 1) for the empirical neural tangent kernel Gram matrix of a trained ResNet50 network on CIFAR-10 with $n = 50,000$ datapoints.

of the matrix and handling permutations and diagonal scaling in the LDL decomposition. The right panel of Figure 1 shows one possible block ordering for the algorithm for LDL and Cholesky decomposition: the ordering is optimal in minimizing the number of reads and writes, but it is not unique. Other block processing orderings with the same amount of data access also exist, and these will be further discussed in Appendix D.1. A detailed analysis of the computational complexity and memory/disk data transfers of the algorithm is provided in Appendix E. Implementation of MEMDET algorithm can be found in Listing I.1.

Although the NTK matrices we work with are symmetric and positive definite (SPD), we do not solely rely on Cholesky decomposition, despite it being the most suitable method for SPD matrices. This is because NTK matrices can lose their positive-definiteness with even the smallest perturbations causing small eigenvalues to become negative, such as when converting from 64-bit to 32-bit precision for efficient computation. As a result, the Cholesky decomposition becomes unstable and fails, necessitating the use of LU and LDL decompositions, suitable for more general matrices. We note that the block computations of LU decomposition can become unstable as the matrices deviate from symmetry and positive-definiteness (Demmel et al., 1995), requiring pivoting of the blocks. However, in our empirical study of NTK matrices, we found that LU decomposition works well without block pivoting, though we do consider pivoting within each block.

## 3. Scaling Law for the Determinant

We now turn our attention to the estimation of the log-determinant of Gram matrices corresponding to covariance kernels. Our primary motivating example is the NTK, an architecture-specific kernel associated with deep neural network models. The connections between neural networks and GP kernels are well-known, particularly the now classi-

cal results that networks at initialization induce a GP kernel in the large-width limit, enabling Bayesian inference for these infinite-width networks (Neal, 1996; Lee et al., 2018). Similarly, linearization of the gradient-flow dynamics during the late stages of training leads to the derivation of the NTK, whose infinite-width analogue can be shown to be constant during this training phase (Jacot et al., 2018; Yang, 2020; Yang & Littwin, 2021).

The NTK was first derived in the context of neural networks. However, the quantity is well defined for a more general class of functions. For a continuously differentiable function $f_{\boldsymbol{\theta}} : \mathcal{X} \to \mathbb{R}^d$, where $d$ is the dimensionality of the model's output (e.g., the number of labels in a classification task), we define its (empirical) NTK as

$$\kappa_{\boldsymbol{\theta}}(x, x') := \mathbf{J}_{\boldsymbol{\theta}}\big(f_{\boldsymbol{\theta}}(x)\big)\mathbf{J}_{\boldsymbol{\theta}}\big(f_{\boldsymbol{\theta}}(x')\big)^{\mathsf{T}}, \qquad (2)$$

where $\mathbf{J}_{\boldsymbol{\theta}}(f_{\boldsymbol{\theta}}(x)) \in \mathbb{R}^{d \times p}$ is the Jacobian of the function $f_{\boldsymbol{\theta}}$ with respect to the flattened vector of its parameters $\boldsymbol{\theta} \in \mathbb{R}^p$, evaluated at the point $x$. The assumption of continuous differentiability is often relaxed in practice.

Note that $\kappa_{\boldsymbol{\theta}}(x, x') \in \mathbb{R}^{d \times d}$, so computing the NTK across $n$ datapoints yields a 4th-order tensor of shape $(n, n, d, d)$. For computational purposes, this is typically flattened into a two-dimensional block matrix of size $nd \times nd$, where each $(i, j)$-block corresponds to the $d \times d$ matrix $\kappa(x_i, x_j)$.

The scaling laws we derive apply to kernel families satisfying decay conditions on the eigenvalues of an associated integral operator (see Appendix F.1 for a precise statement drawn from Li et al. (2023)). This class of kernels includes NTKs, as shown by Bietti & Mairal (2019); Bietti & Bach (2021); Lai et al. (2023); see Figure 2.

### 3.1. Neural Tangent Kernels and Scaling Laws

Let $\kappa : \mathcal{X} \times \mathcal{X} \to \mathbb{R}^{d \times d}$ be a positive-definite kernel. For a sequence of inputs $\{x_i\}_{i=1}^{\infty} \subset \mathcal{X}$ and for each $n \in \mathbb{N}$, let

$\mathbf{K}_n := (\kappa(x_i, x_j))_{i,j=1}^n$ be the corresponding Gram matrix for the first $n$ inputs, which is a matrix of size $nd \times nd$. Our objective is to estimate $\mathsf{logdet}(\mathbf{K}_n)$ for large $n$, using computations involving only smaller values of $n$. Recall that $f$ is a Gaussian process with mean function $\mu : \mathcal{X} \to \mathbb{R}^d$ and covariance kernel $\kappa : \mathcal{X} \times \mathcal{X} \to \mathbb{R}^{d \times d}$, denoted $f \sim \mathcal{GP}(\mu, \kappa)$, if for every $x_1, \ldots, x_n \in \mathcal{X}$ and $d \geq 1$, $(f(x_i))_{i=1}^n \sim \mathcal{N}((\mu(x_i))_{i=1}^n, (\kappa(x_i, x_j))_{i,j=1}^n)$. Our results are founded on the following lemma, which we prove in Appendix F.2.

**Lemma 1.** *Let $f : \mathcal{X} \to \mathbb{R}^d$ be a zero-mean vector-valued $m$-dimensional Gaussian process with covariance kernel $\kappa$. For each $n \geq 2$, let*

$$E(n) := \mathbb{E}[d^{-\frac{1}{2}} \|f(x_n)\|^2 \mid f(x_i) = 0 \, for \, i = 1, \ldots, n-1],$$

*denote the mean-squared error of fitting the $f$ to the zero function using $x_1, \ldots, x_{n-1}$. Then*

$$\frac{\mathsf{pdet}(\mathbf{K}_n)}{\mathsf{pdet}(\mathbf{K}_{n-1})} \leq \left(\frac{d}{r}\right)^{r/2} E(n)^r,$$

*where $r$ is the rank of $\mathrm{Cov}(f(x_n) \mid f(x_i) = 0 \, for \, i = 1, \ldots, n-1)$. In the case where $\mathrm{rank}(\mathbf{K}_n) - \mathrm{rank}(\mathbf{K}_{n-1}) = d$, this reduces to*

$$\frac{\mathsf{pdet}(\mathbf{K}_n)}{\mathsf{pdet}(\mathbf{K}_{n-1})} \leq E(n)^d, \qquad for \, any \, n > 1,$$

*with equality if $d = 1$.*

Lemma 1 is particularly interesting since it highlights a connection between the determinants of Gram matrices and error curves for GPs. This bounds the effect of adding or removing a datapoint on the determinant in terms of the prior variance of a corresponding GP. Previous theoretical (see Appendix F.1) and empirical studies (Spigler et al., 2020; Bahri et al., 2024; Li et al., 2023; Barzilai & Shamir, 2024) have established a power law relationship of the form $E(n) = \Theta(n^{-\xi})$ as $n \to \infty$ for some $\xi > 0$. In view of this, and invoking Lemma 1, we propose the following scaling law for determinants, demonstrated in Figure 2.

**Assumption 1** (SCALING LAW). Assume there exists a constant $C > 0$ and exponent $\nu > 0$ such that as $n \to \infty$,

$$\frac{\det(\mathbf{K}_n)}{\det(\mathbf{K}_{n-1})} = \frac{C}{n^\nu}[1 + o_p(1)]. \tag{3}$$

This assumption allows for accurate estimation of the log-determinants of interest, and an appropriate law of large numbers. However, to construct corresponding confidence intervals, we will need to assume some properties of the error term appearing in Assumption 1. We further impose the mild assumptions of stationarity and bounded variance, demonstrated in Figure 3.

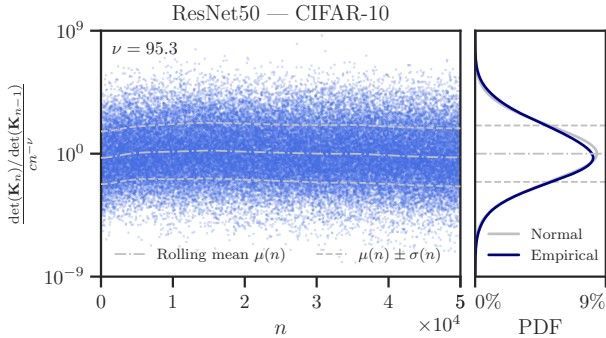

*Figure 3.* Demonstration of the stationarity and second moment behavior (Assumption 2) of the corresponding logarithmic process.

**Assumption 2** (STATIONARITY). Assume there exists a constants $C > 0$ and exponent $\nu > 0$ such that the process $\delta_n$ satisfying

$$\delta_{n-1} := \log\left(\frac{\det(\mathbf{K}_n)}{\det(\mathbf{K}_{n-1})}\right) - \log(Cn^{-\nu}), \quad n = 2, 3, \ldots,$$

is stationary, ergodic, has finite second moment ($\mathbb{E}[\delta_n^2] < +\infty$), and $\mathbb{E}[\delta_n | \delta_1, \ldots, \delta_{n-1}] = 0$.

Under Assumption 1 and Assumption 2, we derive an expression for the asymptotic behavior of the normalized log-determinants. The proof is given in Appendix F.2.

**Proposition 1.** *For larger numbers of inputs, letting*

$$L_n := n^{-1}\mathsf{logdet}(\mathbf{K}_n)$$
$$\hat{L}_n := L_1 + \left(1 - \frac{1}{n}\right)c_0 - \nu\frac{\log(n!)}{n},$$

*a law of large numbers (LLN) and central limit theorem (CLT) hold for the log-determinants $L_n$:*

- (LLN) *Under Assumption 1, there exist constants $c_0, \nu > 0$ such that as $n \to \infty$,*

$$L_n = \hat{L}_n + o_p(1). \tag{4}$$

- (CLT) *Under Assumption 2, there exist constants $c_0, \nu, \sigma > 0$ such that as $n \to \infty$,*

$$\frac{n}{\sqrt{n-1}}(L_n - \hat{L}_n) \xrightarrow{\mathcal{D}} \mathcal{N}(0, \sigma^2). \tag{5}$$

We remark that $\log(n!)$ is $\Theta(n\log(n))$, so the normalized log-determinant $L_n$ is $\Theta_p(\log(n))$ via (4).

### 3.2. FLODANCE

As (4) is linear in the unknown parameters $c_0, \nu$, these parameters can be estimated by linear regression on a pre-computed sequence $(L_1, \ldots, L_n)$.[1] This sequence can be

---

[1] This is generally a more stable regression problem than estimating $C$ and $\nu$ directly from (3).

computed using a single pass of MEMDET on a subsample of the Gram matrix. After performing linear regression, by extrapolating to larger $n$, the normalized log-determinants of larger NTK Gram matrices can be estimated. The ordering of the data is arbitrary, but it will affect the output of the regression task. We refer to this method as the **F**actorial-based **Lo**g-**D**eterminant **A**nalysis and **N**umerical **C**urve **E**stimation procedure, or FLODANCE. To improve performance, we allow for a non-asymptotic correction to the exponent $\nu$ as $\nu(n) \coloneqq \nu_0 + \sum_{i=1}^{q} \nu_i n^{-i}$ (in practice, we find $q \leq 10$ works well). In light of (5) and discussion in Appendix F.3, for large $n$, we expect that approximately

$$y_n = c_0 x_{n,0} + \sum_{i=1}^{q+1} \nu_{i-1} x_{n,i} + \epsilon_n, \quad \epsilon_n \sim \mathcal{N}(0, \sigma^2), \quad (6)$$

where $y_n \coloneqq \frac{n}{\sqrt{n-1}}(L_n - L_1)$, $x_{n,0} \coloneqq \sqrt{n-1}$, and $x_{n,i} \coloneqq \frac{-\log(n!)}{n^{i-1}\sqrt{n-1}}$ for $i = 1, ..., m+1$ are the covariates. The corresponding numerical procedure is presented in Algorithm 1 and the implementation can be found in Listing I.2.

In practice, we also observe that a burn-in period may be required to obtain accurate estimates of $c_0$ and the $\nu_i$ that appear in Proposition 1. Better performance was often achieved in our experiments by discarding the early determinant samples, effectively replacing the $L_1$ term appearing in $y_n$ with different constants $L_{n_0}$, for a burn-in of length $n_0 - 1$. This seems to be due to the sudden emergence of very small eigenvalues that shift the model fit, and constitutes a consistent phenomenon that warrants further investigation. We found that when needed, the burn-in required was always less than 500 terms, verified by cross-validation.

## 4. Numerical Experiments

We now evaluate the accuracy of the FLODANCE algorithm for estimating log-determinants on large-scale problems of interest. The test problems that we consider are NTK matrices corresponding to common deep learning models: ResNet9, ResNet18, and ResNet50 (He et al., 2016) trained on the CIFAR-10 dataset (Krizhevsky, 2009), and MobileNet (Howard et al., 2017) trained on the MNIST dataset (LeCun et al., 1998). Our experiments are split into two sections: first, the dataset size is reduced in order to enable the matrices to fit into memory on a consumer device, in Section 4.1; and then larger subsamples and full datasets are considered, in Section 4.2.[2] As a baseline, we employ MEMDET to compute the relevant quantities, where accuracy is limited only by numerical precision. Detailed runtime and performance diagnostics for MEMDET are pro-

---

[2]Experiments in this section were conducted on a desktop-class device with an AMD Ryzen®7 5800X processor, NVIDIA RTX 3080, and 64GB RAM.

---

**Algorithm 1:** FLODANCE: Factorial-based Log-Determinant Analysis and Numerical Curve Estimation

---

**Input** : Precomputed partial NTK Gram matrix $\mathbf{K}_{n_s}$ of size $m_s \times m_s$ where $m_s = n_s d$,
Model's output dimension $d$,
Total number of datapoints $n$,
Data subsample size $1 < n_s \leq n$,
Burn-in length $1 \leq n_0 < n_s$,
Number of terms in the Laurent series $q$

**Output** : Estimated normalized log-determinant $\hat{L}_n$ of $\mathbf{K}_n$ of size $m \times m$ where $m = nd$

*// $\mathbf{K}_{n_s[:k,:k]}$ is the $k \times k$ principal sub-matrix of $\mathbf{K}_{n_s}$*

**1** Run Algorithm D.2 on $\mathbf{K}_{n_s}$ to obtain $(\ell_k)_{k=1}^{m_s}$ where $\ell_k \leftarrow \mathsf{logabsdet}(\mathbf{K}_{n_s[:k,:k]})$

*// Normalize and record every $d$-th entry.*

**2** Obtain $(L_j)_{j=n_0}^{n_s}$ for $L_j \leftarrow j^{-1}\ell_{m_j}$ and $m_j \leftarrow jd$.

*// Define design matrix $\mathbf{X} \in \mathbf{R}^{(n_s - n_0) \times (q+2)}$ and response vector $\boldsymbol{y} \in \mathbb{R}^{n_s - n_0}$*

**3** **for** $j = 1$ **to** $n_s - n_0$ **do**
**4**    $n_j \leftarrow n_0 + j$
**5**    $y_j \leftarrow n_j(n_j - 1)^{-\frac{1}{2}}(L_{n_j} - L_{n_0})$
**6**    $X_{j,1} \leftarrow (n_j - 1)^{\frac{1}{2}}$
**7**    **for** $i = 1$ **to** $q + 1$ **do**
**8**      $X_{j,i+1} \leftarrow -(n_j - 1)^{-\frac{1}{2}} n_j^{-i+1} \log\Gamma(n_j + 1)$
     *// The Log-gamma function computes $\log(n_j!)$.*

*// Estimate regression coefficients $\boldsymbol{\beta}$ in $\boldsymbol{y} = \mathbf{X}\boldsymbol{\beta} + \epsilon$*

**9** $\boldsymbol{\beta} \leftarrow (\mathbf{X}^\intercal \mathbf{X})^{-1} \mathbf{X}^\intercal \boldsymbol{y}$

*// Estimate $L_n$ at larger value of $n$*

**10** $(c_0, \nu_0, ..., \nu_q) \leftarrow \boldsymbol{\beta}$
**11** $\hat{L}_n \leftarrow$
   $L_{n_0} + c_0(1 - n^{-1}) - \sum_{i=1}^{q+1} \nu_{i-1} n^{-i} \log\Gamma(n+1)$

**12** **return** $\hat{L}_n$

---

vided in Appendix E. Additional experiments evaluating the performance of FLODANCE appear in Appendix G, including its robustness (Appendix G.1) and its application to Matérn kernel Gram matrices (Appendix G.2).

### 4.1. Smaller Data Sets

In order to demonstrate the scaling laws for ratios of successive determinants of NTK matrices, a ResNet50 model was trained on a subset of 1000 datapoints from the CIFAR-10 dataset with $d = 10$ classes. The resulting matrix $\mathbf{K}_{1000}$ is of size $10,000 \times 10,000$. In Figure 2, we plot exact ratios $\det(\mathbf{K}_n)/\det(\mathbf{K}_{n-1})$, with the blue line

*Table 2.* Comparison of approximations of the log-determinant $\hat{\ell}_n$ with the exact computation $\ell_n$ obtained in 64-bit floating-point precision (first row). Values represent average percentage relative errors over five trained networks, with standard deviations in parentheses. Bold values indicate the closest approximation, with the next-best underlined. For the corresponding compute times, see Table H.2.

| Quantity | Model | Configuration | ResNet9 | ResNet9 | ResNet18 | MobileNet |
|---|---|---|---|---|---|---|
| | **Dataset** | | CIFAR-10 | CIFAR-10 | CIFAR-10 | MNIST |
| | **Subsample Size** | | $n = 1000$ | $n = 2500$ | $n = 1000$ | $n = 2500$ |
| $\ell_n$ | Direct Computation (64-bit) | (*Reference*) | 76538 (203) | 181377 (649) | 65630 (842) | $-183962$ (7869) |
| Relative Error $\frac{\|\hat{\ell}_n - \ell_n\|}{\ell_n}$ | Direct Computation (16-bit) | | 12.41% (0.12) | 17.05% (0.13) | 14.00% (0.24) | 66.97% (2.13) |
| | Direct Computation (32-bit) | | 3.67% (0.06) | 6.77% (0.08) | 5.25% (0.09) | **14.27%** (0.95) |
| | SLQ | | 81.51% (0.16) | 80.89% (0.24) | 101.03% (1.64) | 84.52% (1.51) |
| | Block Diagonal | | 76.49% (0.12) | 75.15% (0.16) | 92.76% (1.55) | 112.55% (1.22) |
| | Pseudo NTK | | 118.35% (0.10) | 122.35% (0.27) | 122.95% (0.25) | 75.32% (1.04) |
| | FLODANCE | $n_0 = 1, \quad n_s = 50$ | 7.75% (0.77) | 11.27% (1.10) | 12.19% (0.30) | 36.41% (2.53) |
| | FLODANCE | $n_0 = 1, \quad n_s = 100$ | 5.61% (0.32) | 8.54% (0.63) | 8.09% (0.68) | 35.51% (1.46) |
| | FLODANCE | $n_0 = 300, n_s = 500$ | **1.34%** (0.11) | **1.37%** (0.14) | **2.9%** (0.81) | 23.19% (1.76) |

representing the line of best fit under the scaling law Assumption 1.

For baseline comparison to existing techniques, we compared FLODANCE to approximations of both the matrices themselves and their log-determinants. For the matrix approximations, we consider a block-diagonal approximation ignoring between-data correlations, as well as the pseudo-NTK matrix studied in (Mohamadi et al., 2023). As an approximate log-determinant technique, we consider stochastic Lanczos quadrature, often regarded as the current state-of-the-art for large-scale log-determinant estimation (Gardner et al., 2018). We also compare exact methods across 16-, 32-, and 64-bit (treated as exact) floating-point precision, to assess the accuracy of our extrapolation against memory-saving mixed-precision calculations.

To this end, ResNet9 was trained on a subsets of 1000 and 2500 images from CIFAR-10, and ResNet18 on 1000 datapoints. MobileNet was also trained on a 2500 image subset of the MNIST dataset with $d = 10$ classes. A comparison of the different methods is presented in Table 2. We see that all existing NTK and log-determinant approximation techniques perform poorly when compared with 16- and 32-bit mixed-precision calculations. On the other hand, the FLODANCE estimates that contained a burn-in phase consistently either outperformed the mixed-precision approximations or were competitive. When no burn-in was used, FLODANCE still outperformed the approximation methods. This suggests that at this scale, the error in the scaling law approximation is *less* than the mixed-precision errors discussed in Section 2.1. Extrapolating determinants based on expected behavior can bypass numerical issues at scale. Comparisons for ResNet9 trained on 2500 images from the CIFAR-10 dataset are visualized in Figure 4. FLODANCE estimates consistently outperform the competing methods.

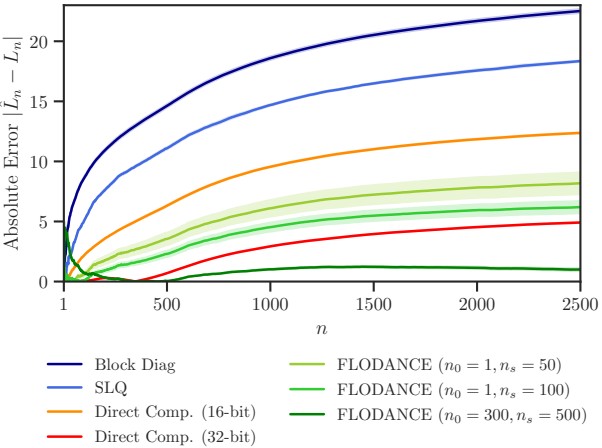

*Figure 4.* Comparison of log-determinant accuracy for NTKs of ResNet9 trained on CIFAR-10, measured by absolute error, across a variety of approximation techniques for matrices of different sizes. Means across five trained networks are displayed, with shaded regions depicting one standard deviation.

### 4.2. Larger Data Sets

In our next experiment, we evaluate NTK matrices at an unprecedented scale, where exact determinant computation has not been previously reported. Due to memory constraints, these matrices cannot be stored explicitly, making MEMDET essential for obtaining ground truth values.

We consider two large-scale NTK matrices: $\mathbf{K}_{50,000}$, a dense matrix of size $500,000 \times 500,000$, for ResNet50 trained on CIFAR-10 with $n = 50,000$ and $d = 10$, and an identical-sized NTK for ResNet9. At this scale, computing the full matrix in double precision poses a formidable challenge. To our knowledge, this is the first time an NTK

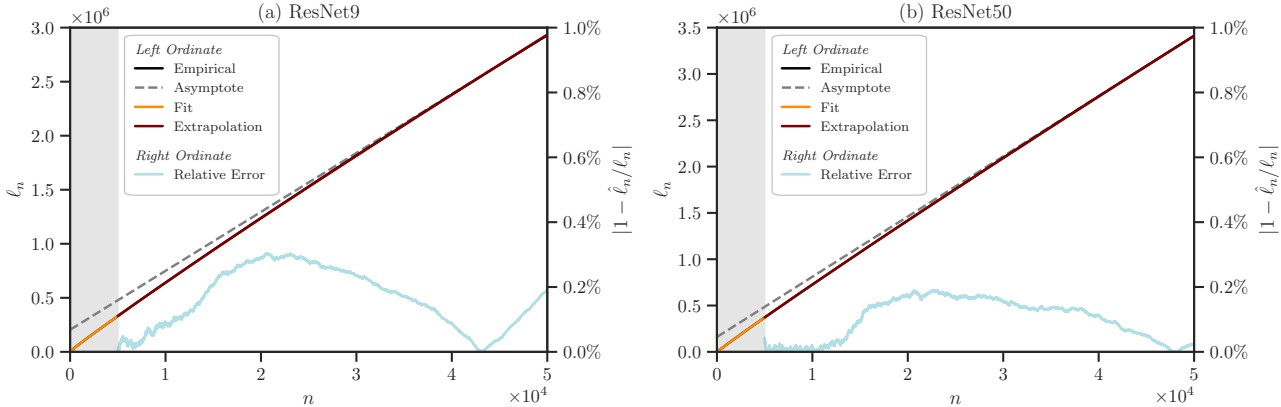

*Figure 5.* Log-determinant $\ell_n$ for $n = 1, \ldots, 50,000$, corresponding to $m \times m$ NTK submatrices where $m = nd$ and $d = 10$, from 64-bit NTK matrices of ResNet9 (a) and ResNet50 (b) trained on CIFAR-10 with 50,000 datapoints. Values are computed using MEMDET (Algorithm D.2) with LDL decomposition (black curves, overlaid by colored curves). FLODANCE (Algorithm 1) is fitted in a small region (shaded gray) and extrapolated over a much larger interval. The yellow curve in the interval $(n_0, n_s) = (10^2, 5 \times 10^3)$ represents the fit, while the red curve in $(n_s, n) = (5 \times 10^3, 5 \times 10^4)$ shows the extrapolation. The blue curves, corresponding to the right axis in each panel shows the relative error, reaching impressive 0.2% in (a) and 0.02% in (b).

matrix of this size—at full precision and over the entire CIFAR-10 dataset—has been computed, with the matrix itself requiring 2 TB of memory (see Appendix C.1). This computation was carried out on an NVIDIA Grace Hopper GH200 GPU over 244 hours for ResNet50.

Having established this benchmark, we now evaluate the accuracy of FLODANCE at this scale. As shown in Figure 5, FLODANCE with $q = 6$, $n_0 = 100$, and $n_s = 5000$ achieves an absolute error of just 0.2% for $\hat{\ell}_{50,000}$ on ResNet9, reducing computation time by a factor of $(n/n_s)^3 = 1000$. Similarly, for ResNet50, FLODANCE with $q = 4$, $n_0 = 100$, and $n_s = 5000$ achieves an absolute error of 0.02% with the same speedup. In contrast, SLQ exhibited a relative error of 55% (see Table 1, also Listing I.3 for implementation). Given their poor performance on smaller datasets, pseudo-NTK and block-diagonal approximations are omitted.

This experiment represents the first exact computation of an NTK determinant at this scale, establishing a new benchmark for large-scale log-determinant estimation. Additional experiments are provided in Appendix G, with Figure G.1 demonstrating the accuracy of the global FLODANCE fit, Figure G.2 illustrating sensitivity to subsample choice, and Figure G.3 examining sensitivity to subsample size.

## 5. Conclusion

The calculation of large matrix log-determinants is a commonly encountered but often avoided problem in statistical and machine learning applications at scale. Several techniques have previously been proposed to circumvent ex-

plicit computation, typically relying on stochastic approximations. However, in many problems, the sheer size of the matrices and their ill-conditioned nature, make not only approximation a difficult task, but forming the matrix itself to provide a baseline becomes computationally intractable. We have addressed this problem on two fronts. On the one hand, we defined MEMDET, a memory-constrained algorithm for log-determinant computation, with different versions for general, symmetric, and symmetric positive-definite matrices. On the other hand, we derived neural scaling laws for large kernel matrices, and we introduced FLODANCE, a procedure for accurate extrapolation of log-determinants from small subsets of the data. The high speed and accuracy of our methods open the door for routine computation of interpolating information criteria and related diagnostic tools to enable principled model selection within deep learning frameworks (Hodgkinson et al., 2023b).

The ability to accurately compute and estimate matrices of this size further provides fascinating insights into the behavior of the NTKs that we considered in our experiments, which treated square matrices of the size up to 500,000. Further, the memory constrained algorithms we described can be applied to other classes of matrices (Nguyen & Vu, 2014; Cai et al., 2015), where they can be expected to unlock similar insights into their scaling behavior. In terms of further computational tools, we are developing techniques to extrapolate more refined spectral information about large matrices from small sub-blocks (Ameli et al., 2025). Methods for blockwise decompositions of large Jacobian matrices would also circumvent the need to explicitly calculate $\mathbf{J}\mathbf{J}^\mathsf{T}$, enabling higher resolution understanding of their behavior.

## Acknowledgments

LH is supported by the Australian Research Council through a Discovery Early Career Researcher Award (DE240100144). FR is partially supported by the Australian Research Council through the Industrial Transformation Training Centre for Information Resilience (IC200100022). MWM acknowledges partial support from DARPA, DOE, NSF, and ONR.

## Impact Statement

This paper presents work whose goal is to advance the field of Machine Learning. There are many potential societal consequences of our work, none which we feel must be specifically highlighted here.

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

# Appendices

## Contents

## A. Nomenclature

We use boldface lowercase letters for vectors, boldface upper case letters for matrices, and normal face letters for scalars, including the components of vectors and matrices. Table A.1 summarizes the main symbols and notations used throughout the paper, organized by context.

*Table A.1.* Common notations used throughout the manuscript.

| Context | Symbol | Description | Example Value |
|---|---|---|---|
| Dataset | $n$ | Number of data points (e.g., images) | 50,000 for CIFAR-10 |
| | $d$ | Number of model outputs (e.g., classes, labels) | 10 for CIFAR-10 |
| | $m$ | Size of NTK matrix is $m \times m$ where $m = nd$ | 500,000 for CIFAR-10 |
| MEMDET | $n_b$ | Number of row/column blocks (i.e., $n_b^2$ blocks in total) | e.g., 16 |
| | $b$ | Block size; typically $b \approx m/n_b$, for $b \times b$ submatrices | e.g., 31,250 |
| | $c$ | Available memory capacity (in bytes) | |
| | $\beta$ | Precision (in bytes) of a floating-point number | |
| FLODANCE | $n_s$ | Number of sampled data points from $n$ | e.g., 2000 |
| | $m_s$ | Size of sampled NTK matrix $m_s = n_s d$ | e.g., 20,000 |
| | $n_0$ | Start of fitting interval $[n_0, n_s]$ | e.g., 100 |
| | $q$ | Truncation order of Laurent series | e.g., 3 |
| | $c_0, \nu_0, ..., \nu_q$ | Regression coefficients | |
| SLQ | $l$ | Lanczos iterations (Krylov subspace size) | e.g., 100 |
| | $s$ | Number of Monte Carlo samples | e.g., 100 |
| Variables | $\mathbf{K}_n$ | NTK matrix with $n$ data points (matrix of size $m$) | |
| | $\mathbf{K}_{n_s}$ | Sampled NTK matrix with $n_s$ data points (matrix of size $m_s$) | |
| | $\ell_n$ | Log-determinant of NTK with $n$ data points (matrix of size $m$) | |
| | $\hat{\ell}_n$ | Estimated log-determinant | |
| | $L_n$ | Normalized log-determinant $L_n = n^{-1}\ell_n$ | |
| | $\boldsymbol{\theta}$ | Vectorization of neural network parameters | |
| | $p$ | Dimension of $\boldsymbol{\theta}$ | |
| Functions | pdet | Pseudo-determinant (product of nonzero eigenvalues) | |
| | logabsdet | Natural logarithm of the absolute value of determinant | |

## B. Related Works in Numerical Linear Algebra

The study of block decomposition methods in numerical linear algebra has a long history. Classical texts such as Golub & Van Loan (2013) and Dongarra et al. (1998b) provide foundational discussions on block LU, block Cholesky, and LDL decompositions, detailing their computational advantages and numerical properties. These methods have been widely used to improve computational efficiency, particularly in high-performance computing (HPC) settings, where recursive block LU (Golub & Van Loan, 2013, Section 3.2.11), parallel LU (Golub & Van Loan, 2013, Section 3.6), and block Cholesky (Golub & Van Loan, 2013, Section 4.2.9) play a central role in large-scale matrix computations.

Beyond theoretical foundations, numerous works have focused on efficient implementations of block factorizations, particularly for parallel architectures. Stark & Beris (1992) optimized block LU decomposition for hierarchical distributed memory, aiming to improve data locality while maintaining parallel efficiency. Dongarra et al. (1979) pioneered high-performance implementations of block factorizations, laying the groundwork for modern HPC systems. More recent studies, such as Galoppo et al. (2005) and Barrachina et al. (2008), have extended block LU methods to GPU-based environments, leveraging parallelism but still assuming that intermediate submatrices fit in memory. While these approaches optimize performance in parallel settings, they do not address the challenge of computing factorizations when the full matrix size far exceeds available RAM. Traditional block methods typically assume that at least some intermediate submatrices can reside in memory, whereas our method (MEMDET) explicitly operates under constrained memory settings, using an out-of-core hierarchical block processing approach.

To address the issue of matrices exceeding main memory capacity, Dongarra et al. (1998a) introduced concepts for parallel out-of-core LU factorization, focusing on efficient data movement between disk and memory. While their work demonstrates how out-of-core computations can be applied to LU factorization, their approach does not extend to log-determinant computations or hierarchical block-wise processing. Similarly, studies on many-core architectures (Venetis & Gao, 2009) and hierarchical memory-aware LU factorizations (Demmel et al., 1993) have improved computational efficiency, but

Table C.1. Memory requirements (for various floating-point precisions) to store empirical NTK matrices of common datasets. The memory is computed as $(nd)^2\beta$, where $n$ is the training set size (second column), $d$ is the number of classes (third column), and $\beta$ is the number of bytes per floating-point value.

| Dataset | Training Set | Classes | Matrix Size | | |
| --- | --- | --- | --- | --- | --- |
| | | | float16 | float32 | float64 |
| CIFAR-10 | 50,000 | 10 | 0.5 TB | 1.0 TB | 2.0 TB |
| MNIST | 60,000 | 10 | 0.72 TB | 1.5 TB | 2.9 TB |
| SVHN | 73,257 | 10 | 1.1 TB | 2.2 TB | 4.2 TB |
| ImageNet-1k | 1,281,167 | 1000 | 3,282,778 TB | 6,565,556 TB | 13,131,111 TB |

Table C.2. Estimated compute time (in hours using an NVIDIA H100 GPU) for NTK matrix computation.

| Dataset | Model | Compute Time (hrs) | | |
| --- | --- | --- | --- | --- |
| | | float16 | float32 | float64 |
| MNIST | MobileNet | 6 | 25 | 50 |
| CIFAR-10 | ResNet9 | 6 | 24 | 70 |
| | ResNet18 | 14 | 63 | 65 |
| | ResNet50 | 37 | 177 | 297 |
| | ResNet101 | 107 | 442 | 1178 |

none have been designed specifically for computing log-determinants under extreme memory constraints, making our approach distinct.

## C. Memory and Computation Challenges of NTK Matrices

### C.1. Storage and Compute Requirements

The empirical NTK serves as a motivating example throughout this work, as it encapsulates key computational challenges associated with large-scale matrix operations. Several software packages have been developed to compute NTK Gram matrices for various neural architectures using automatic differentiation frameworks (Novak et al., 2022; Engel et al., 2022). However, the full formation of these matrices remains computationally prohibitive, even on common benchmark datasets.

Table C.1 presents the storage requirements for NTK matrices corresponding to various datasets, highlighting their enormous size. For instance, even CIFAR-10 requires terabytes of storage, while ImageNet-1k exceeds *exabytes*, making full NTK computation infeasible for most practical applications. Despite its theoretical importance, the NTK Gram matrix is rarely used as a practical tool, with approximations often employed to mitigate computational and memory constraints. Minibatching is one common strategy, and batch-wise NTK approximations have been explored for model selection (Immer et al., 2023). Yet, extending these estimates to full datasets remains an open challenge. Alternative approximation techniques (Mohamadi et al., 2023) have been proposed, but their convergence is only guaranteed in spectral norm, limiting their ability to capture the full spectrum of the NTK. In contrast, the log-determinant—a key quantity in this work—encodes information from the entire eigenvalue distribution, making its computation particularly demanding.

Beyond storage limitations, the computation time for NTK matrices also presents a major challenge. Table C.2 provides estimated compute times for NTK formation across various models and floating-point precisions on an NVIDIA H100 GPU. Even for relatively small datasets like CIFAR-10, NTK computation is expensive, with higher-precision calculations significantly increasing runtime. For large architectures such as ResNet101, double-precision NTK computation can require over a thousand hours, making exact evaluations impractical without algorithmic improvements like those introduced in this work.

### C.2. Data Precisions in Our Computational Pipeline

Our computations follow a multi-stage pipeline, with each stage involving distinct data precision formats and practical constraints:

1. *Model Training.* All neural networks (e.g., ResNet9, ResNet50) were trained using 32-bit precision, which is the default and standard practice in most deep learning frameworks such as PyTorch.

2. *NTK Matrix Computation.* The NTK matrix is computed from the trained model and stored in various precisions (e.g., 16-bit, 32-bit, and 64-bit, from the same pre-trained model). The "precision" of the NTK matrix, as referred to throughout the paper, reflects the compute and storage format at this stage. Due to the high cost of forming these matrices, it is often tempting or necessary to compute and store them in lower precisions. Our low-precision experiments highlight the pitfalls of mixed-precision in these cases as per Section 2.1, regardless of the downstream use case.

3. *Log-Determinant Computation.* Regardless of how the NTK matrix was computed and stored (16-bit, 32-bit, or 64-bit), all log-determinant computations were performed in 64-bit precision across all methods (e.g., MEMDET, SLQ, FLODANCE). This represents a "best-case" mixed-precision setup.

Since MEMDET entirely eliminates memory requirement barriers, it became practical to perform high-precision computations (e.g., 64-bit in stage 3) even on large matrices—thus mitigating common concerns about the overhead associated with higher-precision formats.

## D. Implementation of MEMDET Algorithm

The pseudo-code of the MEMDET algorithm is given in Algorithm D.1 (generic matrices), Algorithm D.2 (symmetric), and Algorithm D.3 (symmetric positive-definite), each computing the log-determinant of a matrix $\mathbf{M}$. The log-determinants of leading principal submatrices of a matrix $\mathbf{M}$ (possibly after permutation, depending on the decomposition) can also be readily computed. For example, for symmetric positive-definite matrices, the Cholesky decomposition $\mathbf{M} = \mathbf{L}\mathbf{L}^{\mathsf{T}}$, with lower-triangular $\mathbf{L}$, gives $\mathsf{logdet}(\mathbf{M}_{[:k,:k]}) = 2\sum_{i=1}^{k}\log(L_{ii})$, where $\mathbf{M}_{[:k,:k]}$ is the $k \times k$ leading principal submatrix.

The memory requirements of MEMDET are determined by a user-defined parameter, allowing it to run on any system regardless of available memory. For an $m \times m$ matrix, the algorithm partitions the data into an $n_b \times n_b$ grid of blocks, each of size $b \times b$, where $b = 1 + \lfloor (m-1)/n_b \rfloor$. The computation requires either 3 or 4 concurrent blocks in memory: for $n_b = 2$, only 3 blocks are needed, requiring $3b^2\beta$ bytes, while for $n_b > 2$, 4 blocks are required, increasing the memory usage to $4b^2\beta$ bytes, where $\beta$ is the number of bytes per floating point.

Given a maximum memory limit $c$ (in bytes), the optimal number of blocks $n_b$ is determined by the parameter $r = m\sqrt{\beta/c}$, with the following selection criteria:

- If $r \leq 1$, the entire matrix fits in memory, so $n_b = 1$.

- If $r \leq \frac{2}{\sqrt{3}}$, three blocks fit in memory, so $n_b = 2$.

- Otherwise, $n_b = \lceil 2r \rceil$.

### D.1. Optimal Sequence of Processing of Blocks

It is important to select an ordering of the blocks to minimize data transfer between disk and memory. The order in which the block $\mathbf{M}_{ij}$ is processed should be chosen to minimize the reading of the blocks $\mathbf{B}$ (corresponding to the index $j$) and $\mathbf{C}$ (corresponding to the index $i$). Ideally, from processing one block to the next, one should update only one of the matrices $\mathbf{B}$ or $\mathbf{C}$, but not both, to reuse one of the blocks already loaded in memory. We formulate this problem of finding the optimal sequence of blocks as follows for the case of LDL/Cholesky decomposition at the $k$-th stage of the algorithm. The case for LU decomposition can be formulated similarly.

Let $G(V, E)$ denote a complete undirected graph with vertices $V \coloneqq \{k+1, \ldots, n_b\}$, where $E$ is the set of all possible edges $e = (v, u)$ between the vertices $u, v \in V$. Each vertex in $V$ represents the event of loading one of the blocks $\mathbf{B} \leftarrow \mathbf{M}_{kj}$ or

---

**Algorithm D.1:** MEMDET: Constrained-Memory Comp. of Log-Det (Case I: *Generic* Matrix)

---

    **Input**   : Matrix $\mathbf{M}$ of size $m \times m$,                     *// stored on disk, may not be loaded on memory*
                   Maximum memory $c$ in bytes
    **Output** : $\ell$: logarithm of the absolute value of the determinant (logabsdet) of $\mathbf{M}$,
                   $\sigma$: sign of the determinant of $\mathbf{M}$

**1**   $r \leftarrow m\sqrt{\beta/c}$                                               *// $\beta$: number of bytes per floating-point*

**2**   **if** $r \le 1$ **then** $n_b \leftarrow 1$                    *// $n_b$: number of row/column blocks, making $n_b \times n_b$ grid of blocks.*
**3**   **else if** $r \le 2/\sqrt{3}$ **then** $n_b \leftarrow 2$
**4**   **else** $n_b \leftarrow \lceil 2r \rceil$

**5**   $b \leftarrow 1 + \lfloor (m-1)/n_b \rfloor$                               *// Size of each block is at most $b \times b$*
**6**   $\ell \leftarrow 0$                                       *// Accumulates log-abs-determinant of diagonal blocks*
**7**   $\sigma \leftarrow 1$                                       *// Keeps track of the parity of matrix*

    *// Allocate memory for block matrices*
**8**   Allocate memory for $b \times b$ matrix $\mathbf{A}$
**9**   **if** $n_b > 1$ **then** Allocate memory for $b \times b$ matrices $\mathbf{B}, \mathbf{C}$
**10**   **if** $n_b > 2$ **then** Allocate memory for $b \times b$ matrix $\mathbf{S}$

    *// Create scratchpad space on disk, large enough to store $n_b(n_b - 1) - 1$ blocks*
**11**   **if** $n_b > 2$ **then** Allocate empty file of the size $(m(m-b) - b^2)\beta$ bytes

    *// Recursive iterations over diagonal blocks*
**12**   **for** $k = 1$ **to** $n_b$ **do**
**13**      **if** $k = 1$ **then** $\mathbf{A} \leftarrow \mathbf{M}_{kk}$                           *// Load from input array on disk*
**14**      $\mathbf{A} \leftarrow \mathbf{PLU}$                  *// In-place LU decomposition with pivoting (written to $\mathbf{A}$)*
**15**      $\ell \leftarrow \ell + \mathsf{logabsdet}(\mathbf{U})$
**16**      $\sigma \leftarrow \sigma \, \mathsf{sgn}(\mathbf{P}) \, \mathsf{sgn}(\mathbf{U})$

**17**      **if** $k < n_b$ **then**
         *// Iterate over row of blocks from bottom upward*
**18**          **for** $i = n_b$ **to** $k+1$ **step** $-1$ **do**
**19**              $\mathbf{C} \leftarrow \mathbf{M}_{ik}^{\mathsf{T}}$       *// Load from disk (from input array if $k = 1$ or from scratchpad if $k > 1$)*
**20**              $\mathbf{C} \leftarrow \mathbf{U}^{-\mathsf{T}}\mathbf{C}$                       *// Solve upper triangular system in-place*

             *// Iterate over column of blocks in alternating directions per row*
**21**              **if** $i - k$ *is even* **then** $(j_{\text{start}}, j_{\text{end}}) \leftarrow (k+1, n_b)$
**22**              **else** $(j_{\text{start}}, j_{\text{end}}) \leftarrow (n_b, k+1)$

**23**              **for** $j = j_{\text{start}}$ **to** $j_{\text{end}}$ **step** $(-1)^{i-k}$ **do**
                 *// Load $\mathbf{B}$ from disk (input array if $k = 1$ and $i = n_b$, otherwise from scratchpad)*
**24**                  **if** $i = n_b$ **or** $j \ne j_{\text{start}}$ **then** $\mathbf{B} \leftarrow \mathbf{M}_{kj}$
**25**                  **if** $i = n_b$ **then**
**26**                      $\mathbf{B} \leftarrow \mathbf{L}^{-1}\mathbf{P}^{\mathsf{T}}\mathbf{B}$                  *// Solve lower triangular system in-place*
**27**                      **if** $n_b - k > 2$ **or** $j \ne j_{\text{end}}$ **then** $\mathbf{M}_{kj} \leftarrow \mathbf{B}$      *// Write to disk on scratchpad*

**28**                  **if** $i = k+1$ **and** $j = k+1$ **then**
**29**                      $\mathbf{A} \leftarrow \mathbf{M}_{ij}$            *// Load from disk (input array if $k = 1$ or scratchpad if $k > 1$)*
**30**                      $\mathbf{A} \leftarrow \mathbf{A} - \mathbf{C}^{\mathsf{T}}\mathbf{B}$                    *// Compute Schur complement*
**31**                  **else**
**32**                      $\mathbf{S} \leftarrow \mathbf{M}_{ij}$            *// Load from disk (input array if $k = 1$ or scratchpad if $k > 1$)*
**33**                      $\mathbf{S} \leftarrow \mathbf{S} - \mathbf{C}^{\mathsf{T}}\mathbf{B}$                    *// Compute Schur complement*
**34**                      $\mathbf{M}_{ij} \leftarrow \mathbf{S}$                     *// Write to disk on scratchpad*

**35**   **return** $\ell, \sigma$

---

---

**Algorithm D.2:** MEMDET: Constrained-Memory Comp. of Log-Det (Case II: *Symmetric* Matrix)

---

**Input** : Symmetric matrix $\mathbf{M}$ of size $m \times m$,                       *// stored on disk, may not be loaded on memory*
            Maximum memory $c$ in bytes
**Output** : $\boldsymbol{\pi} := (\pi_q)_{q=1}^m$: a permutation of $\{1, \ldots, m\}$            *// permutations induced by LDL decomposition*
          $\boldsymbol{\ell} := (\ell_q)_{q=1}^m$: $\ell_q := \mathsf{logabsdet}(\mathbf{M}_{[\mathcal{I}_q, \mathcal{I}_q]})$ with the index set $\mathcal{I}_q := (\pi_1, \ldots, \pi_q)$       *// $\log |\det(\cdot)|$*
          $\boldsymbol{\sigma} := (\sigma_q)_{q=1}^m$: $\sigma_q := \mathsf{sgn}(\det(\mathbf{M}_{[\mathcal{I}_q, \mathcal{I}_q]}))$                      *// Sign of $\det(\cdot)$*

**1** $r \leftarrow m\sqrt{\beta/c}$                            *// $\beta$: number of bytes per floating-point*

**2 if** $r \leq 1$ **then** $n_b \leftarrow 1$             *// $n_b$: number of row/column blocks, making $n_b \times n_b$ grid of blocks.*
**3 else if** $r \leq 2/\sqrt{3}$ **then** $n_b \leftarrow 2$ **else** $n_b \leftarrow \lceil 2r \rceil$

**4** $b \leftarrow 1 + \lfloor (m-1)/n_b \rfloor$                  *// Size of each block is at most $b \times b$*
**5** Initialize arrays $\boldsymbol{d} \in \mathbb{R}^m$ and $\boldsymbol{\pi} \in \{1, \ldots, m\}^m$        *// Hold diagonals and permutations, respectively*

*// Allocate memory for block matrices*
**6** Allocate memory for $b \times b$ matrix $\mathbf{A}$
**7 if** $n_b > 1$ **then** Allocate memory for $b \times b$ matrices $\mathbf{B}, \mathbf{C}$
**8 if** $n_b > 2$ **then** Allocate memory for $b \times b$ matrix $\mathbf{S}$
**9 if** $n_b > 1$ **then** Define pointers $\mathbf{B}_\star, \mathbf{C}_\star$         *// Used for swapping memory; $(\mathbf{B}_\star, \mathbf{C}_\star)$ will refer to $(\mathbf{B}, \mathbf{C})$ or $(\mathbf{C}, \mathbf{B})$*

*// Create scratchpad space on disk, large enough to store $n_b(n_b + 1)/2 - 4$ blocks*
**10 if** $n_b > 2$ **then** Allocate empty file of the size $(m(m+b)/2 - 4b^2)\beta$ bytes

*// Recursive iterations over diagonal blocks*
**11 for** $k = 1$ **to** $n_b$ **do**
**12**     **if** $k = 1$ **then** $\mathbf{A} \leftarrow \mathbf{M}_{kk}$                 *// Load from input array on disk*
**13**     $\mathbf{A} \leftarrow \mathbf{PLDL}^\mathsf{T}\mathbf{P}^\mathsf{T}$            *// In-place $LDL^T$ decomposition with pivoting (written to $\mathbf{A}$)*
**14**     $\boldsymbol{d}_{[1+(k-1)b:kb]} \leftarrow \mathsf{diag}(\mathbf{D})$           *// Accumulate diagonals of $\mathbf{D}$ to $\boldsymbol{d}$*
**15**     $\boldsymbol{\pi}_{[1+(k-1)b:kb]} \leftarrow (k-1)b + \mathsf{permutation}(\mathbf{P})$        *// Accumulate permutation indices*

**16**     **if** $k < n_b$ **then**
        *// Iterate over column of blocks backward (right to left)*
**17**        **for** $j = n_b$ **to** $k+1$ **step** $-1$ **do**
**18**           **if** $n_b - j$ *is even* **then** $(\mathbf{B}_\star, \mathbf{C}_\star) = (\mathbf{B}, \mathbf{C})$ **else** $(\mathbf{B}_\star, \mathbf{C}_\star) = (\mathbf{C}, \mathbf{B})$      *// swap $\mathbf{B}$ and $\mathbf{C}$ memories*

**19**           **if** $j = n_b$ **then**
**20**              $\mathbf{B}_\star \leftarrow \mathbf{M}_{kj}$          *// Load from disk (input array if $k = 1$ or from scratchpad if $k > 1$)*
**21**              $\mathbf{B}_\star \leftarrow \mathbf{L}^{-1}\mathbf{P}^\mathsf{T}\mathbf{B}_\star$              *// Solve lower-triangular system in-place*

**22**           $\mathbf{C} \leftarrow \mathbf{B}_\star$          *// Deep copy of the memory pointed by $\mathbf{B}_\star$ to memory pointed by $\mathbf{C}_\star$*
**23**           $\mathbf{B}_\star \leftarrow \mathbf{D}^{-1}\mathbf{B}_\star$

          *// Processing order of rows: first process row $j$, then from row $k + 1$ downward to $j - 1$*
**24**           $\mathcal{R} \leftarrow (j, k+1, k+2, \ldots, j-2, j-1)$

**25**           **for** $i = \mathcal{R}(1)$ **to** $\mathcal{R}(j - k)$ **do**
**26**              **if** $i \neq j$ **then** $\mathbf{C}_\star \leftarrow \mathbf{M}_{ki}$        *// Load disk (input array if $k = 1$, $j = n_b$, otherwise scratchpad)*

**27**              **if** $j = n_b$ **then**
**28**                 $\mathbf{C}_\star \leftarrow \mathbf{L}^{-1}\mathbf{P}^\mathsf{T}\mathbf{C}_\star$              *// Solve lower triangular system in-place*
**29**                 **if** $n_b > 2$ **and** $i < j - 1$ **then** $\mathbf{M}_{ki} \leftarrow \mathbf{C}_\star$      *// Write to disk on scratchpad*

**30**              **if** $i = k + 1$ **and** $j = k + 1$ **then**
**31**                 $\mathbf{A} \leftarrow \mathbf{M}_{ij}$           *// Load from disk (input array if $k = 1$ or scratchpad if $k > 1$)*
**32**                 $\mathbf{A} \leftarrow \mathbf{A} - \mathbf{C}_\star^\mathsf{T}\mathbf{B}_\star$               *// Compute Schur complement*

**33**              **else**
**34**                 $\mathbf{S} \leftarrow \mathbf{M}_{ij}$           *// Load from disk (input array if $k = 1$ or scratchpad if $k > 1$)*
**35**                 $\mathbf{S} \leftarrow \mathbf{S} - \mathbf{C}_\star^\mathsf{T}\mathbf{B}_\star$                *// Compute Schur complement*
**36**                 $\mathbf{M}_{ij} \leftarrow \mathbf{S}$            *// Write to disk on scratchpad*

**37** $(\ell_0, \sigma_0) \leftarrow (0, 1)$
**38 for** $q = 1$ **to** $m$ **do** $(\ell_q, \sigma_q) \leftarrow (\ell_{q-1} + \log(|d_q|), \sigma_{q-1}\,\mathsf{sgn}(d_q))$         *// $d_q$ is the $q$-th element of $\boldsymbol{d}$*

**39 return** $\boldsymbol{\pi}, \boldsymbol{\ell}, \boldsymbol{\sigma}$

---

---

**Algorithm D.3:** MEMDET: Constrained-Memory Comp. of Log-Det (Case III: *Symmetric Positive-Definite* Matrix)

---

**Input** : Symmetric positive-definite matrix $\mathbf{M}$ of size $m \times m$,          *// stored on disk, not on memory*
         Maximum memory $c$ in bytes
**Output** : $\boldsymbol{\ell} := (\ell_q)_{q=1}^m : \ell_q := \mathsf{logdet}(\mathbf{M}_{[:q,:q]})$

1   $r \leftarrow m\sqrt{\beta/c}$                      *// $\beta$: number of bytes per floating-point*

2   **if** $r \leq 1$ **then** $n_b \leftarrow 1$          *// $n_b$: number of row/column blocks, making $n_b \times n_b$ grid of blocks.*
3   **else if** $r \leq 2/\sqrt{3}$ **then** $n_b \leftarrow 2$ **else** $n_b \leftarrow \lceil 2r \rceil$

4   $b \leftarrow 1 + \lfloor (m-1)/n_b \rfloor$                *// Size of each block is at most $b \times b$*
5   Initialize array $\boldsymbol{d} \in \mathbb{R}^m$                  *// Holds diagonals*

    *// Allocate memory for block matrices*
6   Allocate memory for $b \times b$ matrix $\mathbf{A}$
7   **if** $n_b > 1$ **then** Allocate memory for $b \times b$ matrices $\mathbf{B}, \mathbf{C}$
8   **if** $n_b > 2$ **then** Allocate memory for $b \times b$ matrix $\mathbf{S}$
9   **if** $n_b > 1$ **then** Define pointers $\mathbf{B}_\star, \mathbf{C}_\star$          *// Used for swapping memory; $(\mathbf{B}_\star, \mathbf{C}_\star)$ will refer to $(\mathbf{B}, \mathbf{C})$ or $(\mathbf{C}, \mathbf{B})$*

    *// Create scratchpad space on disk, large enough to store $n_b(n_b + 1)/2 - 4$ blocks*
10   **if** $n_b > 2$ **then** Allocate empty file of the size $(m(m+b)/2 - 4b^2)\beta$ bytes

    *// Recursive iterations over diagonal blocks*
11   **for** $k = 1$ **to** $n_b$ **do**
12      **if** $k = 1$ **then** $\mathbf{A} \leftarrow \mathbf{M}_{kk}$               *// Load from input array on disk*
13      $\mathbf{A} \leftarrow \mathbf{L}\mathbf{L}^\mathsf{T}$                  *// In-place Cholesky decomposition (written to $\mathbf{A}$)*
14      $\boldsymbol{d}_{[1+(k-1)b:kb]} \leftarrow \mathsf{diag}(\mathbf{L})$          *// Accumulate diagonals of $\mathbf{L}$ to $\boldsymbol{d}$*

15      **if** $k < n_b$ **then**
        *// Iterate over column of blocks backward (right to left)*
16        **for** $j = n_b$ **to** $k+1$ **step** $-1$ **do**
17          **if** $n_b - j$ *is even* **then** $\mathbf{B}_\star = \mathbf{B}$ **else** $\mathbf{B}_\star = \mathbf{C}$      *// Alternate pointer $\mathbf{B}_\star$ to switch between $\mathbf{B}$ and $\mathbf{C}$*

18          **if** $j = n_b$ **then**
19            $\mathbf{B}_\star \leftarrow \mathbf{M}_{kj}$           *// Load from disk (input array if $k = 1$ or from scratchpad if $k > 1$)*
20            $\mathbf{B}_\star \leftarrow \mathbf{L}^{-1}\mathbf{B}_\star$           *// Solve lower-triangular system in-place*

         *// Processing order of rows: first process row $j$, then from row $k+1$ downward to $j-1$*
21          $\mathcal{R} \leftarrow (j, k+1, k+2, \ldots, j-2, j-1)$
22          **for** $i = \mathcal{R}(1)$ **to** $\mathcal{R}(j-k)$ **do**

23            **if** $i = j$ **then** $\mathbf{C}_\star = \mathbf{B}_\star$          *// Shallow copy of pointer $\mathbf{C}_\star$ pointing to $\mathbf{B}_\star$*
24            **else**
25              **if** $n_b - j$ *is even* **then** $\mathbf{C}_\star = \mathbf{C}$ **else** $\mathbf{C}_\star = \mathbf{B}$      *// Alternate pointer $\mathbf{C}_\star$ between $\mathbf{C}$ and $\mathbf{B}$*
26              $\mathbf{C}_\star \leftarrow \mathbf{M}_{ki}$       *// Load $\mathbf{C}_\star$ from disk (input array if $k = 1$, $j = n_b$, otherwise scratchpad)*

27              **if** $j = n_b$ **then**
28                $\mathbf{C}_\star \leftarrow \mathbf{L}^{-1}\mathbf{C}_\star$          *// Solve lower triangular system in-place*
29                **if** $n_b > 2$ **and** $i < j-1$ **then** $\mathbf{M}_{ki} \leftarrow \mathbf{C}_\star$      *// Write to disk on scratchpad*

30            **if** $i = k+1$ **and** $j = k+1$ **then**
31              $\mathbf{A} \leftarrow \mathbf{M}_{ij}$          *// Load from disk (input array if $k = 1$ or scratchpad if $k > 1$)*
32              $\mathbf{A} \leftarrow \mathbf{A} - \mathbf{C}_\star^\mathsf{T}\mathbf{B}_\star$          *// Compute Schur complement*
33            **else**
34              $\mathbf{S} \leftarrow \mathbf{M}_{ij}$          *// Load from disk (input array if $k = 1$ or scratchpad if $k > 1$)*
35              $\mathbf{S} \leftarrow \mathbf{S} - \mathbf{C}_\star^\mathsf{T}\mathbf{B}_\star$          *// Compute Schur complement*
36              $\mathbf{M}_{ij} \leftarrow \mathbf{S}$          *// Write to disk on scratchpad*

37   $\ell_0 \leftarrow 0$
38   **for** $q = 1$ **to** $m$ **do** $\ell_q \leftarrow \ell_{q-1} + 2\log(d_q)$          *// $d_q$ is the $q$-th element of $\boldsymbol{d}$*

39   **return** $\boldsymbol{\ell}$

---

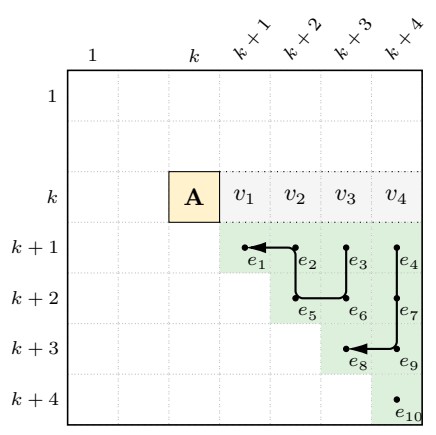 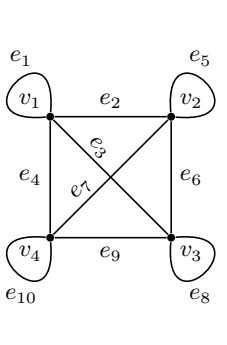 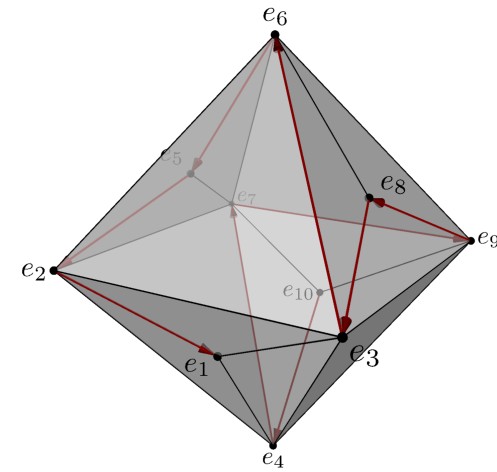

*Figure D.1. Left:* Example of the processing order of blocks for a symmetric matrix at the $k$-th hierarchical step. In this step, to process $\mathbf{S} \leftarrow \mathbf{M}_{ij}$, the memory blocks $\mathbf{B}$ and $\mathbf{C}$ are selected from the set $V = \{v_1, v_2, v_3, v_4\}$. *Middle:* The corresponding complete graph $G(V, E)$. *Right:* The corresponding line graph $L(G)$, with one possible Hamiltonian path highlighted in red, starting from the node $e_{10}$ and ending at the node $e_1$.

$\mathbf{C} \leftarrow \mathbf{M}_{ik}$, $i, j = k + 1, \ldots, n_b$. Each edge in $E$ corresponds to the event of processing the block $\mathbf{M}_{ij}$. At the $k$-th stage of the algorithm, eventually, all blocks $i, j = k + 1, \ldots, n_b$ will be processed, so $E$ consists of all edges of a complete graph, including self-loops, with $|E| = \frac{|V|(|V|+1)}{2}$. To illustrate this concept, consider the example in Figure D.1. The left panel depicts the $k$-th iteration of the algorithm for a symmetric matrix, where the matrices $\mathbf{B}$ and $\mathbf{C}$ have four blocks to choose from the set $V = \{v_1, v_2, v_3, v_4\}$. The corresponding graph $G$ is shown in the middle panel of the figure.

The goal is to select an ordered sequence $(e_p)$, $p = 1, \ldots, |E|$ of edges such that each two consecutive edges $e_p$ and $e_{p+1}$ in the sequence share a common vertex. This ensures that from processing one block to the next, only one of $\mathbf{B}$ or $\mathbf{C}$ is updated, while at least one block is reused from the previous step.

To find such a sequence of edges, we define $L(G)$, the *line graph* of $G$ (also called the *edge-to-vertex dual*), where each vertex of $L(G)$ represents an edge of $G$. Two vertices in $L(G)$ are adjacent if and only if their corresponding edges in $G$ share a vertex. Thus, any *Hamiltonian path* in $L(G)$ yields an ordered edge sequence fulfilling our requirement.

As illustrated in Figure D.1, the right panel depicts the line graph of the given graph shown in the middle panel, with a possible Hamiltonian path highlighted in red. This path directly translates to the processing order of blocks shown in the left panel. Notably, all valid Hamiltonian paths must terminate at the node $e_1$, representing the block $\mathbf{M}_{k+1,k+1}$. This specific end point is crucial as it allows for a seamless transition to the next iteration (i.e., the $k + 1$ iteration) without the need to explicitly load the matrix $\mathbf{A}$, as it would already be available in memory from the last processing block of the $k$-th iteration when $\mathbf{S} \leftarrow \mathbf{M}_{k+1,k+1}$ was processed.

Given that $G$ is complete and therefore Hamiltonian, it follows that its line graph $L(G)$ is also Hamiltonian. This implies the existence of at least one (but possibly many) Hamiltonian paths. Crucially, all Hamiltonian paths in $L(G)$ have the same length. Consequently, any sequence of blocks derived from a Hamiltonian path constitutes an optimal solution to our problem. Thus, the block sequence presented in Figure 1 is equivalent in optimality to any other sequence obtainable from a Hamiltonian path.

The same problem can be formulated for the block LU decomposition, with the modification that $G$ is a complete and balanced bipartite graph $G(V, V, E)$; however, the same logic and conclusion follow.

## E. Complexity and Performance Analysis of MEMDET

### E.1. Computational Complexity

Table E.1 provides a detailed breakdown of the computational complexity of MEMDET for generic matrices (second column, using LU decomposition) and symmetric matrices (third column, using LDL or Cholesky decomposition). The

*Table E.1.* Breakdown of computational complexity for MEMDET. The table presents the number of operations performed and the FLOP count per operation for generic matrices (LU decomposition) and symmetric matrices (LDL or Cholesky decomposition). The last row shows the total complexity, which remains independent of the number of blocks $n_b$.

| Operation | Generic Matrix | | Symmetric (Positive-Definite) Matrix | |
|---|---|---|---|---|
| | Num. Operations | FLOPs per Operation | Num. Operations | FLOPs per Operation |
| Matrix Decomposition | $n_b$ | $\frac{1}{3}b^3 - \frac{1}{2}b^2 + \frac{1}{6}b$ | $n_b$ | $\frac{1}{6}b^3 - \frac{1}{4}b^2 + \frac{1}{12}b$ |
| Solve Lower Triangular System | $\frac{1}{2}n_b^2 - \frac{1}{2}n_b$ | $\frac{1}{2}b^3 - \frac{1}{2}b^2$ | $\frac{1}{2}n_b^2 - \frac{1}{2}n_b$ | $\frac{1}{2}b^3 - \frac{1}{2}b^2$ |
| Solve Upper Triangular System | $\frac{1}{2}n_b^2 - \frac{1}{2}n_b$ | $\frac{1}{2}b^3 - \frac{1}{2}b^2$ | | |
| Full Matrix Multiplication | $\frac{1}{3}n_b^3 - \frac{1}{2}n_b^2 + \frac{1}{6}n_b$ | $b^3$ | $\frac{1}{6}n_b^3 - \frac{1}{2}n_b^2 + \frac{1}{3}n_b$ | $b^3$ |
| Gramian Matrix Multiplication | | | $\frac{1}{2}n_b^2 - \frac{1}{2}n_b$ | $\frac{1}{2}b^3$ |
| Total Complexity | | $\frac{1}{3}m^3 - \frac{1}{2}m^2 + \frac{1}{6}m$ | | $\frac{1}{6}m^3 - \frac{1}{4}m^2 + \frac{1}{12}m$ |

operations are categorized into matrix decomposition, solving triangular systems, and matrix multiplications used to form Schur complements. Each operation's complexity is given in terms of the number of times it is performed and the FLOP count per operation. In this analysis, one FLOP refers to a fused multiply-add (FMA) operation—one multiplication and one addition—as counted by modern GPU benchmarks. The table lists a unified complexity column for symmetric matrices, encompassing both LDL and Cholesky decompositions. While LDL includes additional operations such as row permutations and diagonal scaling via $\mathbf{D}$, these are excluded from the FLOP count due to their negligible cost relative to the leading terms.

The complexity of each operation is given by the number of times it is performed (a function of $n_b$) multiplied by the FLOP count per operation (a function of the block size $b$). Substituting $b = m/n_b$ into these expressions, the total complexity, obtained by summing across all operations, simplifies such that $n_b$ cancels out, as shown in the last row of Table E.1. Thus, the total computational complexity of MEMDET is independent of the number of blocks $n_b$, and is identical to that of conventional factorization algorithms where $n_b = 1$.

Although the total computational cost remains the same, the contribution of individual operations shifts as $n_b$ increases. When $n_b = 1$, the entire computation consists solely of a matrix decomposition. As $n_b$ increases, the decomposition cost decreases while additional operations, such as solving triangular systems and matrix multiplications, account for a larger fraction of the total complexity. At the extreme case of $n_b = m$, the algorithm consists primarily of matrix multiplications. This transition is illustrated in Figure E.1 (left panel), where the contributions of matrix decomposition, triangular system solving, and matrix multiplication are plotted as functions of $n_b$. The total computational complexity, shown as the black curve, remains constant, while the distribution of work among different operations shifts as $n_b$ increases.

### E.2. Data Transfer and Memory Considerations

While the total FLOP count is independent of $n_b$, the number of data transfers between memory and disk increases with the number of blocks. Table E.2 summarizes the number of blocks read from disk to memory and written back to disk, as a function of $n_b$. The actual volume of transferred data is obtained by multiplying the number of blocks by the block size, $b^2\beta$, where $\beta$ represents the number of bytes per floating point. The right panel of Figure E.1 illustrates the total data transfer volume relative to the original matrix size.

For $n_b \leq 2$, the entire computation is performed in memory, avoiding disk I/O and thus requiring no scratchpad space. However, for $n_b > 2$, the computation utilizes scratchpad space, and data transfer overhead increases approximately as $\mathcal{O}(n_b^2)$, meaning that choosing an excessively high $n_b$ introduces unnecessary I/O costs. Despite this, the hierarchical design of MEMDET efficiently schedules block transfers, mitigating excessive data movement.

Table E.3 provides an analysis of the required memory and scratchpad space. The number of blocks stored in memory and on disk is determined by $n_b$, with the actual space usage obtained by multiplying the number of blocks by the block size. By adjusting $n_b$, MEMDET can be configured to run within any given memory constraint, making it adaptable to systems with limited memory.

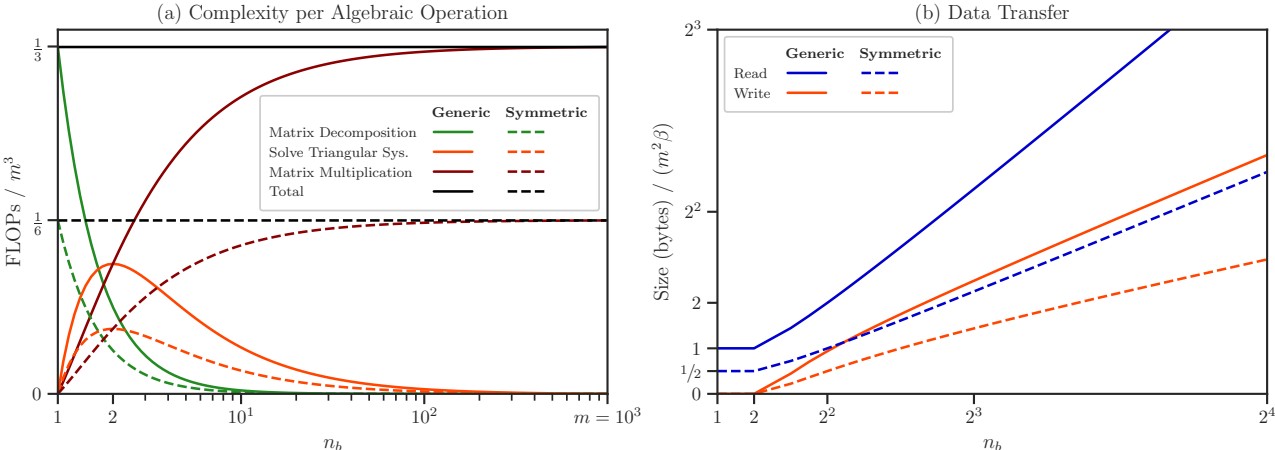

*Figure E.1.* Theoretical computational complexity of MEMDET as a function of the number of blocks $n_b$. The left panel shows the contributions of matrix decomposition, solving triangular systems, and matrix multiplication to the total complexity. The black curve represents the total computational cost, which remains constant, while the colored curves illustrate how the workload shifts across operations as $n_b$ increases. The right panel displays the total data transfer volume (normalized by the original matrix size) for different $n_b$, highlighting the increasing cost of disk I/O as the number of blocks grows.

*Table E.2.* Number of blocks transferred between disk and memory during MEMDET execution. The total data transfer volume is obtained by multiplying the number of transferred blocks by the block size, $b^2\beta$ bytes, where $b = m/n_b$. Read operations occur in all cases, while write operations to the scratchpad are only required for $n_b > 2$.

| Operation | Generic Matrix | | Symmetric Matrix | |
|---|---|---|---|---|
| Read | $\frac{2}{3}n_b^3 - n_b^2 + \frac{4}{3}n_b$ | | $\frac{1}{3}n_b^3 - \frac{1}{2}n_b^2 + \frac{7}{6}n_b$ | |
| Write | $\begin{cases} 0, & n_b \leq 2 \\ \frac{1}{3}n_b^3 - \frac{4}{3}n_b - 1, & n_b > 2 \end{cases}$ | | $\begin{cases} 0, & n_b \leq 2 \\ \frac{1}{6}n_b^3 + \frac{1}{2}n_b^2 - \frac{11}{3}n_b + 4, & n_b > 2 \end{cases}$ | |

*Table E.3.* Number of concurrent blocks that must be allocated in memory (first row) and the total number of blocks allocated on disk (second row) during MEMDET execution. The total required memory and disk space are obtained by multiplying the number of allocated blocks by the block size, $b^2\beta$ bytes. While the number of concurrent memory-resident blocks remains fixed, the total number of blocks allocated on disk increases with $n_b > 2$.

| Hardware | Generic Matrix | | Symmetric Matrix | |
|---|---|---|---|---|
| Memory | 3 or 4 | | 3 or 4 | |
| Scratchpad | $\begin{cases} 0, & n_b \leq 2 \\ n_b^2 - n_b - 1, & n_b > 2 \end{cases}$ | | $\begin{cases} 0, & n_b \leq 2 \\ \frac{1}{2}n_b^2 + \frac{1}{2}n_b - 4, & n_b > 2 \end{cases}$ | |

### E.3. Empirical Performance Evaluation

To validate the theoretical complexity and memory analysis, we conducted empirical evaluations on SPD matrices of various sizes, ranging from $m = 2^{10}$ to $2^{16}$. The largest matrix tested was chosen to match the memory capacity of a 64 GB system, allowing for a direct comparison between MEMDET and conventional algorithms. For each matrix size, the algorithm was executed with different numbers of blocks, $n_b = 1, 2, \ldots, 8$, where $n_b = 1$ corresponds to a standard full-matrix decomposition with the entire matrix loaded into memory. Each experiment was repeated 10 times, and the mean and standard deviation of the profiling measures are reported.

Figure E.2 presents the experimental results. The left panel shows peak memory allocation, measured using a memory profiling tool, which precisely matches the theoretical predictions. As expected, when $n_b = 1$, the required memory

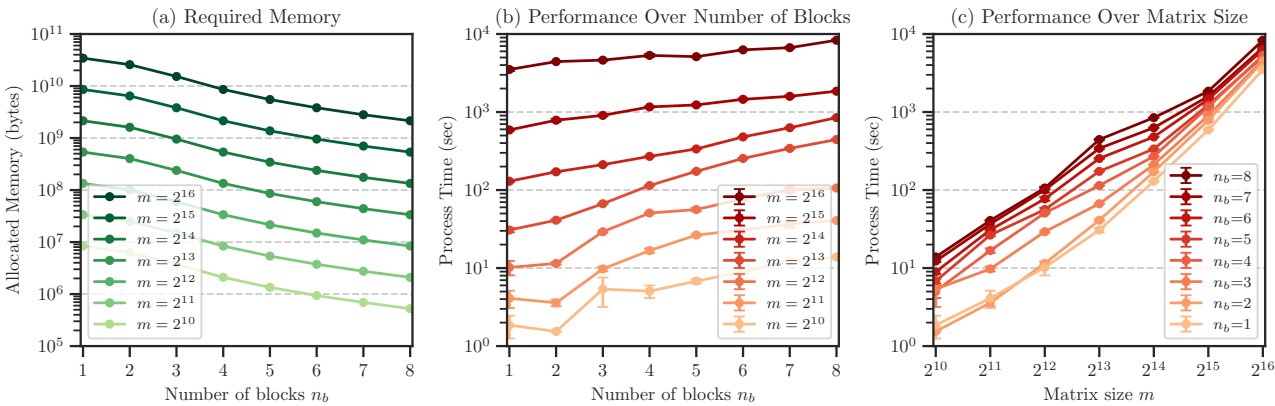

*Figure E.2.* Peak memory allocation (*a*) and CPU processing time (*b*, *c*) for MEMDET on symmetric positive-definite matrices of size $m = 2^{10}, \ldots, 2^{16}$, using Algorithm D.3. The matrices were processed using an $n_b \times n_b$ grid of matrix blocks, where $n_b = 1, 2, \ldots, 8$.

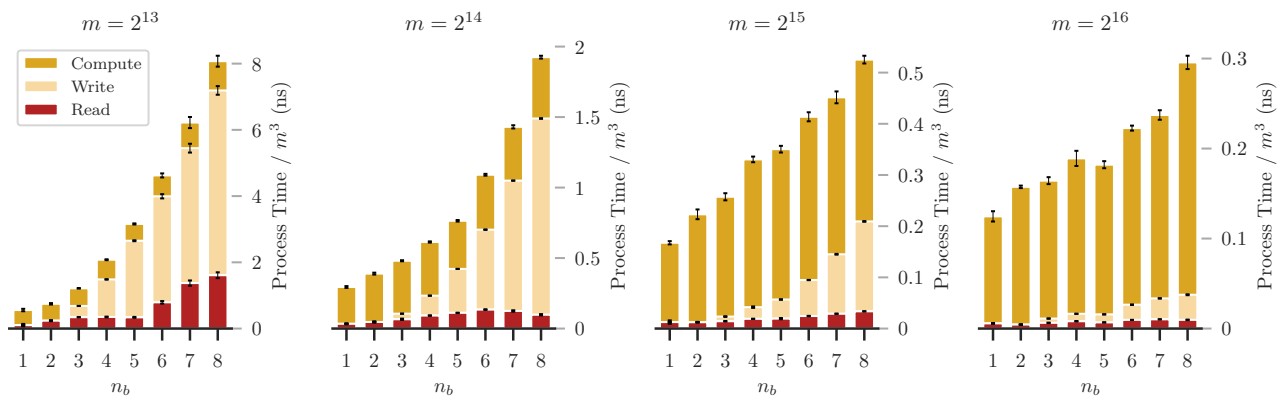

*Figure E.3.* Breakdown of MEMDET runtime (using Algorithm D.3) into computation and data transfer times, normalized by $m^3$, for $m = 2^{13}, \ldots, 2^{16}$. The total process time consists of reading from disk to memory (maroon), writing from memory to disk (light tan), and CPU computation (ochre).

equals the original matrix size, while increasing $n_b$ reduces the memory footprint. The middle and right panels display the measured process time as functions of $n_b$ and $m$, respectively. At large $m$, the difference in process time across varying $n_b$ diminishes, indicating that the increase in data transfer cost does not significantly impact overall runtime.

To further analyze this effect, Figure E.3 decomposes the process time into computation time and data transfer time. At small $m$, the runtime is dominated by disk I/O, but as $m$ increases, computation time becomes the dominant factor. This confirms that for sufficiently large matrices, the performance of MEMDET approaches that of conventional in-memory methods.

### E.4. Concluding Remarks of Performance Analysis

MEMDET maintains the same computational complexity as conventional factorization methods while distributing computations across blocks. The total FLOP count remains unchanged, but increasing $n_b$ shifts the workload between operations (i.e., from matrix decompositions to matrix multiplications). However, increasing $n_b$ also increases data transfer overhead, requiring a balance between reducing memory usage and minimizing disk I/O.

The scheduling design of block operations optimizes memory usage while limiting unnecessary data movement, ensuring that MEMDET remains efficient under constrained memory conditions. By allowing users to specify a memory limit, MEMDET enables the processing of arbitrarily large matrices on systems with any limited memory size.

For large-scale applications, where conventional methods exceed memory capacity, MEMDET provides a practical alterna-

*Table E.4.* Relative error of MEMDET (using $n_b = 8$ blocks) and Numpy's `eigh` with respect to NumPy's `slogdet` (used as baseline) across different models, dataset sizes, and precision formats.

| Quantity | Model | ResNet9 | ResNet9 | ResNet18 | MobileNet |
|---|---|---|---|---|---|
| | **Dataset**
**Subsample Size** | CIFAR-10
$n = 1000$ | CIFAR-10
$n = 2500$ | CIFAR-10
$n = 1000$ | MNIST
$n = 2500$ |
| Rel. Error | MEMDET (16-bit) | $2.2 \times 10^{-15}$ | $5.4 \times 10^{-15}$ | $5.4 \times 10^{-14}$ | $8.7 \times 10^{-16}$ |
| | MEMDET (32-bit) | $4.2 \times 10^{-12}$ | $4.3 \times 10^{-10}$ | $1.7 \times 10^{-11}$ | $8.4 \times 10^{-12}$ |
| | MEMDET (64-bit) | $1.4 \times 10^{-8}$ | $5.4 \times 10^{-10}$ | $1.8 \times 10^{-9}$ | $1.7 \times 10^{-6}$ |
| | `eigh` (16-bit) | $1.6 \times 10^{-14}$ | $1.7 \times 10^{-14}$ | $2.8 \times 10^{-14}$ | $1.8 \times 10^{-14}$ |
| | `eigh` (32-bit) | $5.0 \times 10^{-10}$ | $1.1 \times 10^{-9}$ | $9.2 \times 10^{-12}$ | $4.5 \times 10^{-12}$ |
| | `eigh` (64-bit) | $6.1 \times 10^{-8}$ | $7.5 \times 10^{-9}$ | $1.4 \times 10^{-9}$ | $1.0 \times 10^{-5}$ |

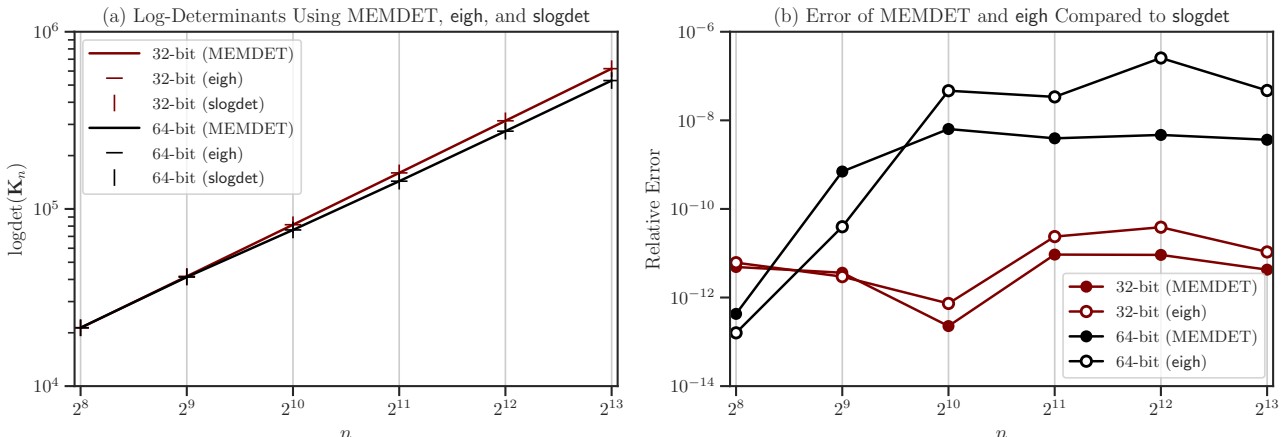

*Figure E.4.* Log-determinants of growing NTK submatrices from ResNet9, computed in both 32-bit (red) and 64-bit (black) formats. Each submatrix has size $m = nd$ with $d = 10$. (a) Comparison between MEMDET, `slogdet`, and `eigh` for each input matrix. (b) Relative error of MEMDET and `eigh` with respect to `slogdet`. Regardless of the matrix precision, all log-determinant computations are performed in 64-bit precision, and errors remain well below $10^{-7}$.

tive. The experiments confirm that while data transfer overhead exists, it does not significantly impact runtime at large $m$, making MEMDET a viable solution for large-matrix computations on standard hardware.

### E.5. Computational Accuracy of MEMDET

We validate the numerical accuracy of MEMDET by comparing its log-determinant output across various precision formats against two standard in-memory methods: `numpy.linalg.slogdet` and `numpy.linalg.eigh` (from which the log-determinant is computed as the sum of the logarithms of the eigenvalues). Table E.4 shows that MEMDET matches both methods with relative errors between $10^{-8}$ and $10^{-16}$, well within the margin of numerical agreement between the baselines themselves.

To assess behavior at larger scales, we compute log-determinants of growing NTK submatrices derived from ResNet9, with the matrices formed in both 32-bit and 64-bit floating-point formats. Figure E.4 confirms that MEMDET remains in tight agreement with `slogdet` and `eigh` across all scales. Notably, discrepancies between the 32-bit and 64-bit curves are attributable solely to differences in the data precision of the underlying input NTK matrices.

Finally, we evaluate the effect of MEMDET's block size parameter $n_b$, which determines the number of memory partitions used during computation. Figure E.5 shows that even for NTK matrices of size up to 100,000, increasing the number of blocks has no measurable impact on accuracy: all results remain within $10^{-16}$ to $10^{-12}$ relative error compared to the full in-memory LDL decomposition.

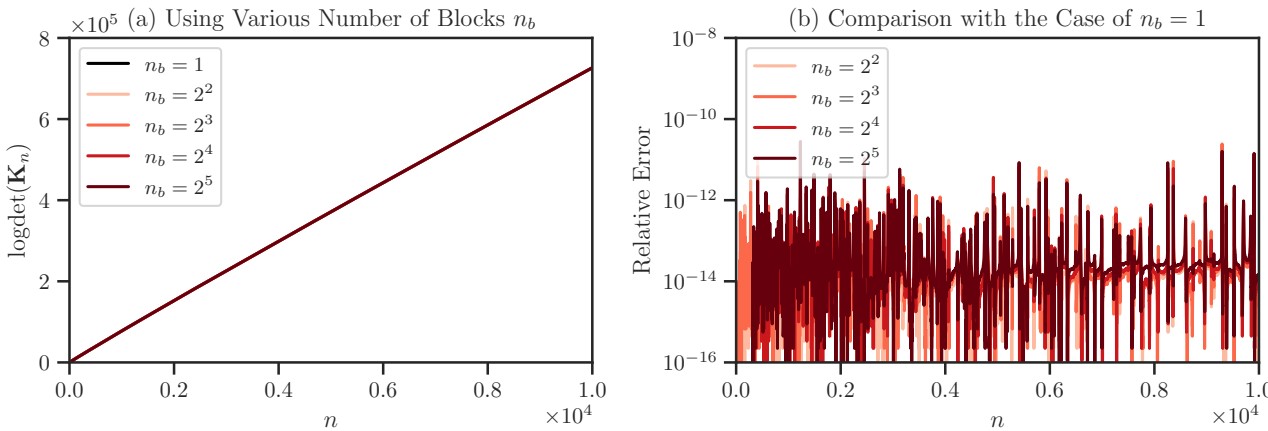

*Figure E.5.* Effect of the number of blocks $n_b$ used in MEMDET. (a) Log-determinants computed for NTK submatrices from ResNet50, with different values of $n_b$. (b) Relative error with respect to the conventional LDL decomposition ($n_b = 1$). All results match to within $10^{-16}$ to $10^{-12}$ accuracy, indicating high numerical stability.

# F. FLODANCE and Scaling Laws

In this section, we collect supporting theoretical content related to the FLODANCE algorithm. Appendix F.1 reviews background material on neural scaling laws. Appendix F.2 provides the proofs of our main lemma and proposition. Finally, Appendix F.3 presents the derivation of the FLODANCE parameterization.

## F.1. Background Material for Neural Scaling Laws

FLODANCE builds on recent theoretical developments in neural scaling laws that characterize generalization error in terms of kernel eigenvalue decay and source smoothness. Here, we review the key assumptions and results from Li et al. (2023) that underpin our bias analysis. These assumptions concern the spectral properties of the kernel, the embedding behavior of the corresponding RKHS, and the regularity of the target function. We restate them below for completeness.

The following result from Li et al. (2023), stated under a noiseless setting consistent with our framework, relies on the following assumptions:

**Assumption F.1** (Eigenvalue Decay). There exists a $\beta > 1$ and constants $c_\beta, C_\beta > 0$ such that

$$c_\beta i^{-\beta} \leq \lambda_i \leq C_\beta i^{-\beta}, \tag{F.1}$$

where the $\lambda_i$ are eigenvalues of the kernel $k : \mathcal{X} \times \mathcal{X} \to \mathbb{R}$ under the decomposition guaranteed by Mercer's Theorem:

$$\kappa(x, x') = \sum_{i=1}^{\infty} \lambda_i e_i(x) e_i(x'). \tag{F.2}$$

We need to define an embedding index associated to certain interpolation spaces that arise as the range of fractional powers of integral operators. In order to do so, we define the integral operator $T : L^2 \to L^2$ that acts as the natural embedding of a RKHS $\mathcal{H}$ associated with our kernel $\kappa$, precomposed with its adjoint. That is, $T$ is the integral operator given by

$$(Tf)(x) = \int_{\mathcal{X}} \kappa(x, x') f(x') d\mu(x'),$$

where $\mu$ is the marginal distribution of $\rho$ on $\mathcal{X}$, where $\rho$ is the source distribution on $(\mathcal{X} \times \mathcal{Y})$ underlying the dataset. The operator $T$ can be decomposed by the spectral theorem of compact self-adjoint operators via

$$T = \sum_{i=1}^{\infty} \lambda_i \langle \cdot, e_i \rangle_{L^2} e_i. \tag{F.3}$$

For $s \geq 0$, this lets us define the fractional powers $T^s : L^2 \to L^2$ of the operator $T$ to satisfy

$$T^s(f) = \sum_{i=1}^{\infty} \lambda_i^s \langle f, e_i \rangle_{L^2} e_i. \tag{F.4}$$

The interpolation space $[\mathcal{H}]^s$ associated to $T^{s/2}$ can then be defined as

$$[\mathcal{H}]^s = \mathrm{range}(T^{s/2}) = \left\{ \sum_{i=1}^{\infty} a_i \lambda_i^{\frac{s}{2}} e_i \,\middle|\, \sum_{i=1}^{\infty} a_i^2 < \infty \right\} \subset L^2. \tag{F.5}$$

We now say that $\mathcal{H}$ has an embedding property of order $\alpha \in (0, 1]$ if $[\mathcal{H}]^\alpha$ can be continuously embedded into $L^\infty$. Define then the operator norm, which has the form (see (Fischer & Steinwart, 2020))

$$\| [\mathcal{H}]^s \hookrightarrow L^\infty \| = \mathrm{ess} \sup_{x \in \mathcal{X}, \mu} \sum_{i=1}^{\infty} \lambda_i^\alpha e_i(x)^2.$$

We now have the following assumption on the embedding index, which is known to be satisfied if the eigenfunctions $e_i$ are uniformly bounded (Steinwart et al., 2009).

**Assumption F.2** (Embedding index). The embedding index $\alpha_0 = 1/\beta$, where $\beta$ is the eigenvalue decay in (F.1), and $\alpha_0$ is defined as

$$\alpha_0 = \inf \{ \alpha : \| [\mathcal{H}]^s \hookrightarrow L^\infty \| = M_\alpha < \infty \}$$

Finally, we have the following assumption on the smoothness of the source function $f_\rho^\star = \mathbb{E}_\rho[y \,|\, x]$, which is a more precise characterization than requiring it to belong to some interpolation space.

**Assumption F.3** (Source condition). There exists an $s > 0$ and a sequence $(a_i)_{i \geq 1}$ for which

$$f_\rho^\star = \sum_{i=1}^{\infty} a_i \lambda_i^{\frac{s}{2}} i^{-\frac{1}{2}} e_i,$$

and $0 < c \leq |a_i| \leq C$ for some constants $c, C$.

These assumptions are required for the following theorem, taken from (Li et al., 2023).

**Theorem F.1.** *Under Assumptions F.1 to F.3, fix $s > 1$ and suppose that $\lambda \asymp n^{-\theta}$ for $\theta \geq \beta$. Then*

$$\mathrm{Bias}^2 = E(n) = \mathcal{O}_{\mathbb{P}}^{\mathrm{poly}}(n^{-\min(s,2)\beta}).$$

## F.2. Proofs of Lemma 1 and Proposition 1

**Proof of Lemma 1.** Fix $\alpha > 0$ and write $\mathbf{K}_n^\alpha = \mathbf{K}_n + \alpha \mathbf{I}_m$ where $\mathbf{I}_m$ is the $m \times m$ identity matrix. In block form, $\mathbf{K}_n^\alpha$ contains $\mathbf{K}_{n-1}^\alpha$ according to

$$\mathbf{K}_n^\alpha = \begin{bmatrix} \mathbf{K}_{n-1}^\alpha & \kappa(\boldsymbol{x}_{n-1}, x_n) \\ \kappa(\boldsymbol{x}_{n-1}, x_n)^\intercal & \kappa(x_n, x_n) + \alpha \mathbf{I}_d \end{bmatrix},$$

where $\kappa(\boldsymbol{x}_{n-1}, x_n) = (\kappa(x_i, x_n))_{i=1}^{n-1} \in \mathbb{R}^{(n-1)d \times d}$. Consequently, since $\mathbf{K}_n^\alpha$ and $\mathbf{K}_{n-1}^\alpha$ are both positive-definite, their determinants differ by the Schur determinant:

$$\det(\mathbf{K}_n^\alpha) = \det(\mathbf{K}_{n-1}^\alpha) \det \left( \kappa(x_n, x_n) + \alpha \mathbf{I}_d - \kappa(\boldsymbol{x}_{n-1}, x_n)^\intercal [\mathbf{K}_{n-1}^\alpha]^{-1} \kappa(\boldsymbol{x}_{n-1}, x_n) \right).$$

Combining the Sylvester rank inequality with Corollary 20 from (Ameli & Shadden, 2023), we take $\alpha \downarrow 0$ and observe that

$$\mathsf{pdet}(\mathbf{K}_n) \leq \mathsf{pdet}(\mathbf{K}_{n-1}) \, \mathsf{pdet}(\mathrm{Cov}(f(x_n) \,|\, f(x_i) = 0 \text{ for } i = 1, \ldots, n-1)). \tag{F.6}$$

This lets us apply the AM-GM inequality and then bound the Frobenius norm in terms of the nuclear norm to obtain

$$\frac{\mathsf{pdet}(\mathbf{K}_n)}{\mathsf{pdet}(\mathbf{K}_{n-1})} \leq \mathsf{pdet}(\mathrm{Cov}(f(x_n) \mid f(x_i) = 0 \text{ for } i = 1, \ldots, n-1)) = \prod_{j=1}^{r} \lambda_j \leq r^{-\frac{r}{2}} \left( \sum_{j=1}^{r} \lambda_j^2 \right)^{\frac{r}{2}}$$

$$\leq r^{-\frac{r}{2}} \mathsf{trace}\left( \mathrm{Cov}(f(x_n) \mid f(x_i) = 0 \text{ for } i = 1, \ldots, n-1)^r \right)$$

$$\leq \left( \frac{d}{r} \right)^{r/2} E(n)^r,$$

where the $\lambda_j$ are the non-zero eigenvalues of the covariance matrix, and $r$ is its rank.

$\square$

**Proof of Proposition 1.** From equation (3),

$$\mathsf{logdet}(\mathbf{K}_n) - \mathsf{logdet}(\mathbf{K}_{n-1}) = \log C - \nu \log n + \log[1 + o_p(1)]$$
$$= \log C - \nu \log n + o_p(1), \tag{F.7}$$

and so

$$\mathsf{logdet}(\mathbf{K}_n) - \mathsf{logdet}(\mathbf{K}_1) = (n-1) \log C - \nu \log(n!) + o_p(n).$$

Letting $c_0 := \log C - \mathsf{logdet}(\mathbf{K}_1)$ and dividing by $n$ implies the first result. For the second result, we replace the $o_p(1)$ term in (F.7) with $\delta_{n-1}$. Consequently,

$$\frac{n}{\sqrt{n-1}} \left[ L_n - L_1 - \left( 1 - \frac{1}{n} \right) c_0 + \nu \frac{\log(n!)}{n} \right] = (n-1)^{-1/2} \sum_{i=1}^{n-1} \delta_i,$$

and from (Billingsley, 1961), $(n-1)^{-1/2} \sum_{i=1}^{n-1} \delta_i$ converges weakly to a normal random variable with zero mean and variance $\sigma^2 = \mathbb{E}[\delta_1^2]$. $\square$

### F.3. Derivation of FLODANCE Parameterization

Here we derive the equation (6), which leads to the numerical procedure in Algorithm 1. To show that the coefficients $c_0$ and $\nu_0, \ldots, \nu_q$ can be obtained using standard linear regression procedures, it is necessary to determine the form of the covariates $x_{n,i}$. Letting $\nu_n := \nu_0 + \sum_{i=1}^{q} \nu_i n^{-i}$, Proposition 1 implies that, asymptotically in $n$,

$$L_n = L_1 + \left( 1 - \frac{1}{n} \right) c_0 - \left( \nu_0 + \sum_{i=1}^{q} \frac{\nu_i}{n^i} \right) \frac{\log(n!)}{n} + \frac{\sqrt{n-1}}{n} \epsilon_n, \qquad \epsilon_n \overset{\text{iid}}{\sim} \mathcal{N}(0, \sigma^2),$$

for some $c_0, \nu_0, \ldots, \nu_q$ and $\sigma > 0$. Rearranging, there is

$$\frac{n}{\sqrt{n-1}} (L_n - L_1) = \frac{n-1}{\sqrt{n-1}} c_0 - \left( \nu_0 + \sum_{i=1}^{q} \frac{\nu_i}{n^i} \right) \frac{\log(n!)}{\sqrt{n-1}} + \epsilon_n,$$

Equivalently, the above relation can be recast as a linear regression problem:

$$y_n = c_0 x_{n,0} + \sum_{i=1}^{q+1} \nu_{i-1} x_{n,i} + \epsilon_n.$$

where $c_0, \nu_0, \ldots, \nu_q$ are the regression coefficients for the covariates $x_{n,i}$ defined as

$$x_{n,i} = \begin{cases} \sqrt{n-1}, & i = 0, \\ \dfrac{\log(n!)}{n^{i-1}\sqrt{n-1}}, & i = 1, \ldots q. \end{cases}$$

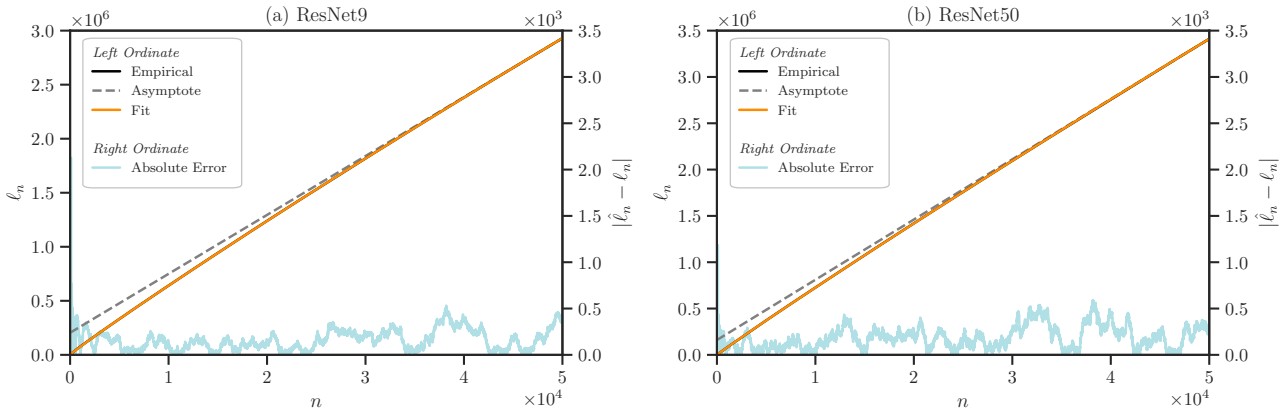

*Figure G.1.* Log-determinant $\ell_n$ for $n = 1, \ldots, 50{,}000$, corresponding to $m \times m$ NTK submatrices where $m = nd$ and $d = 10$, from 64-bit NTK matrices of ResNet9 (*a*) and ResNet50 (*b*) trained on CIFAR-10 with 50,000 datapoints. Values are computed using MEMDET (Algorithm D.2) with LDL decomposition (black curves, overlaid by colored curves). The orange curves represent theoretical fits based on the parametrization derived in Algorithm 1. Fitting is performed globally over the entire interval $(n_0, n) = (1, 5 \times 10^4)$, demonstrating the accuracy of the theoretical model. The blue curve, corresponding to the right axis (scaled to one-thousandth of the left axis), shows the absolute error.

The target variable for regression is given by

$$y_n = \frac{n}{\sqrt{n-1}}(L_n - L_1).$$

We note that in numerical implementations, the term $\log(n!)$ should be evaluated using the log-gamma function: $\log \Gamma(n + 1) = \log(n!)$.

# G. Further Empirical Analysis of FLODANCE

This section investigates the accuracy, robustness, and generality of FLODANCE through two complementary studies. Appendix G.1 revisits NTK matrices, quantifying the method's sensitivity to fitting-interval length and to random subsampling. Appendix G.2 then applies FLODANCE to a multi-output Gaussian process with a Matérn kernel, demonstrating that the same scaling-law machinery applies well beyond neural tangent kernels.

## G.1. Sensitivity and Robustness on NTK Matrices

**Global Fit of FLODANCE on NTK Matrices.**  To validate the theoretical parameterization given in Appendix F.3, we compute the log-determinants of NTK submatrices of size $1, \ldots, n$ from ResNet9 and ResNet50 trained on CIFAR-10, with $n = 50{,}000$. The log-determinants are obtained using MEMDET (Algorithm D.2) with LDL decomposition. Figure G.1 presents the empirical results: black curves (largely overlaid by orange) show the computed log-determinants, while the orange curves show the theoretical fits derived from the parameterization in Algorithm 1. The fitting is performed globally over the entire interval, demonstrating the accuracy of the theoretical model in capturing the log-determinant behavior. This global fit complements the extrapolation-based application shown earlier in Figure 5, where FLODANCE is trained on a small subset and extrapolated to the full range. The near-perfect overlap between the curves highlights the quality of the fit, with errors remaining below 0.05% for most of the interval.

**Uncertainty Under Subsampling.**  To assess the robustness of FLODANCE predictions, we evaluate how subsampling affects log-determinant estimates. In Figure G.2, we generate 15 independent subsamples of NTK matrices of size $m = nd$, with $d = 10$ and $n = 10{,}000$, from ResNet50 trained on CIFAR-10. Panel (a) shows the exact log-determinants $\ell_n$ computed using MEMDET for each subsample. The black curve denotes the ensemble mean, while the gray shading around the mean (barely visible) indicates the standard deviation across subsamples. The red curve (right axis of panel) shows the normalized standard deviation, which remains below 0.1% throughout, indicating remarkable stability of the log-determinants under subsampling.

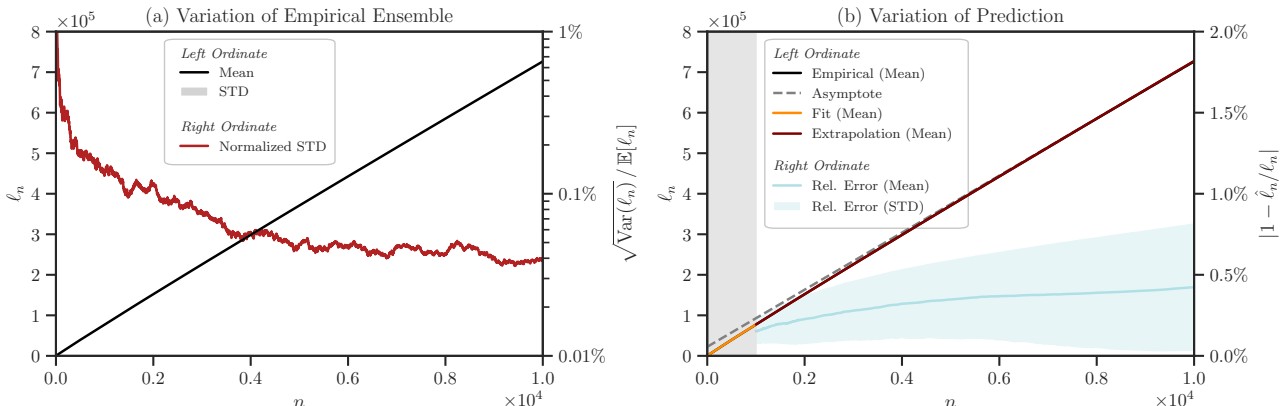

*Figure G.2.* Sensitivity of log-determinant estimates to subsampling variation. (a) Exact log-determinants $\ell_n$ of 15 randomly subsampled NTK matrices of size $m \times m$, $m = nd$, with number of classes $d = 10$ and data points $n = 10{,}000$, computed using MEMDET on ResNet50 trained on CIFAR-10. The black curve denotes the mean, and the shaded gray (barely visible) shows the standard deviation across subsamples. The right ordinate shows the normalized standard deviation (red), which remains below 0.1%. (b) Predicted log-determinants using FLODANCE fitted over a small interval $(n_0, n_s) = (1, 10^3)$ (yellow), and extrapolated to $(n_s, n) = (10^3, 10^4)$ (red). The left axis shows the predicted mean; the right axis shows the relative error (blue), with mean error under 1% and variation across ensemble (shaded blue).

Panel (b) evaluates the predictive performance of FLODANCE on the same ensembles, where the model is trained on the interval $(n_0, n_s) = (1, 10^3)$ and extrapolated to $(n_s, n) = (10^3, 10^4)$. The predicted mean log-determinant is shown in red (extrapolated) and yellow (fitted), closely tracking the ensemble mean of the exact log-determinants (black, largely obscured). The right axis displays the relative error: the blue curve denotes the mean, which stays below 0.5%, and the shaded region shows the standard deviation across ensembles, which remains comparably tight. These results confirm that FLODANCE exhibits both accuracy and robustness under random subsampling.

**Effect of Fitting Interval Size.** We investigate the sensitivity of FLODANCE to the choice of the fitting interval size $n_s$, which governs the trade-off between computational cost and extrapolation accuracy. Recall that FLODANCE fits a model to log-determinants computed over the interval $[n_0, n_s]$ and extrapolates to the full range $[n_s, n]$, where $n_0 = 1$ and $n = 50{,}000$ in this experiment. Figure G.3 evaluates this trade-off on NTK matrices from ResNet50 trained on CIFAR-10. As $n_s$ increases, more data is used for fitting (increasing the cost), but less extrapolation is required (increasing the accuracy).

Panel (a) shows the root-mean-square (RMS) error of the fit in the training interval $[n_0, n_s]$, which increases with $n_s$ due to the growing number of points to match. Panel (b) displays the RMS error in the extrapolation region $[n_s, n]$, which decreases as the extrapolation range shrinks. Finally, (c) plots the relative error of the prediction at the endpoint $n = 50{,}000$, which drops sharply from over 50% to under 0.1% as $n_s$ increases. These results illustrate the central design principle of FLODANCE: by choosing a moderate $n_s$, one can achieve accurate predictions while keeping the cost of computing exact log-determinants limited to smaller submatrices.

### G.2. Extension to Matérn Kernel Gaussian Process

Finally, we demonstrate the generality of FLODANCE beyond NTKs by applying it to a multi-output Gaussian process with a Matérn kernel, widely used in spatial statistics due to its tunable smoothness. The isotropic Matérn correlation function of Matérn (1960) (see also Stein (1999, p. 31)) between two spatial points $\boldsymbol{x}, \boldsymbol{x}' \in \mathbb{R}^p$ is given by

$$\rho(\boldsymbol{x}, \boldsymbol{x}' \,|\, \alpha, \nu) = \frac{2^{1-\nu}}{\Gamma(\nu)} \left( \sqrt{2\nu}\, \frac{\|\boldsymbol{x} - \boldsymbol{x}'\|_2}{\alpha} \right)^{\nu} K_{\nu} \left( \sqrt{2\nu}\, \frac{\|\boldsymbol{x} - \boldsymbol{x}'\|_2}{\alpha} \right),$$

where $\Gamma(\cdot)$ is the Gamma function and $K_{\nu}(\cdot)$ is the modified Bessel function of the second kind of order $\nu$ (Abramowitz & Stegun, 1964, Section 9.6). The hyperparameter $\nu$ modulates the smoothness of the underlying random process, and the hyperparameter $\alpha > 0$ is the correlation scale of the kernel. We construct the multi-output covariance using the

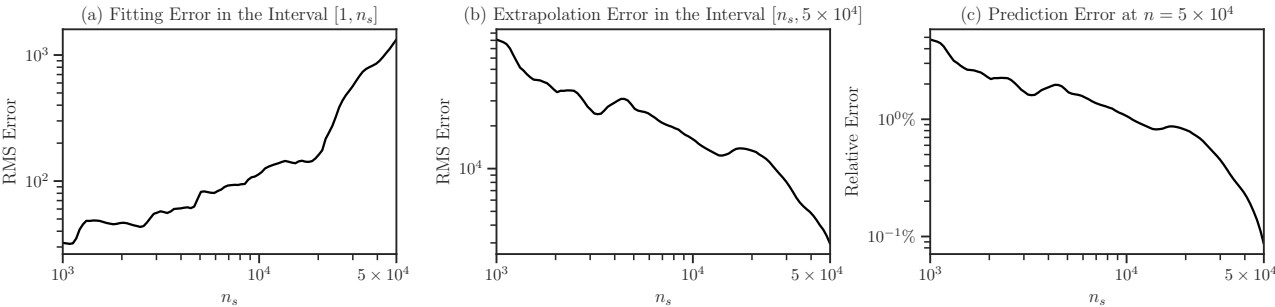

*Figure G.3.* Sensitivity of FLODANCE to the choice of fitting interval size $n_s$. Based on ResNet50 trained on the full CIFAR-10 dataset with $n = 5 \times 10^4$ data points and $d = 10$ classes, resulting in NTK matrices of size $m = nd = 10^5$. FLODANCE extrapolates log-determinants by fitting a model on submatrices of size $n_0 = 1$ to $n_s$, and extending this fit to larger sizes up to $n$. (a) Root-mean-square error (RMSE) of the fit in the interval $[1, n_s]$, showing increasing fitting error as $n_s$ grows. (b) RMSE of extrapolation in the interval $[n_s, n]$, which decreases with $n_s$. (c) Relative error of predicting the log-determinant at $n = 5 \times 10^4$, again decreasing with $n_s$.

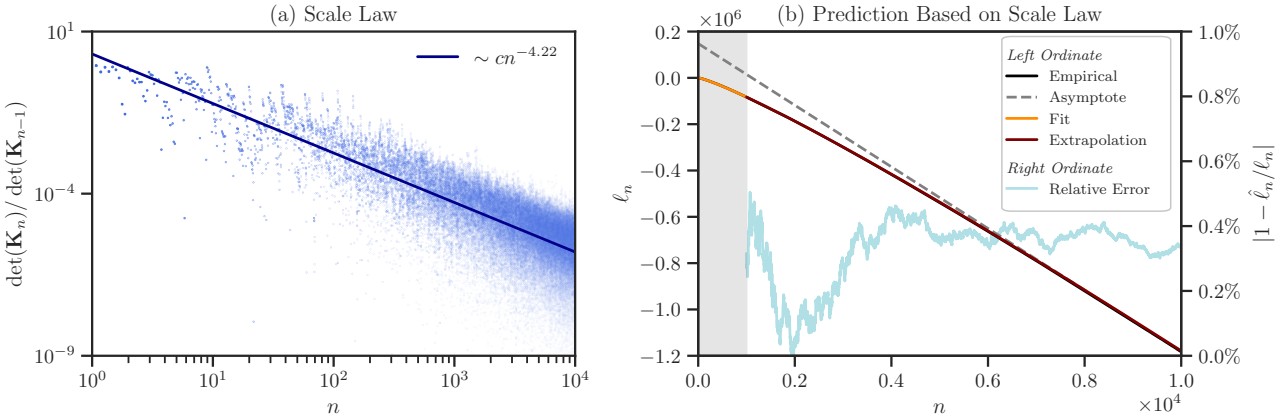

*Figure G.4.* Application of FLODANCE to a multi-output Gaussian process with a Matérn kernel. We generate $n = 10,000$ spatial locations in $\mathbb{R}^2$ and assume a $d = 10$-dimensional output per location, resulting in a covariance matrix of size $m = nd = 100,000$. The covariance structure follows a Matérn kernel with smoothness $\nu = 1.5$ and scale parameter $\alpha = 0.04$, combined with a linear model of coregionalization (LMC) for output covariances. (a) Scale law illustrated by the ratio of successive determinants over increasing submatrix sizes. (b) Log-determinant prediction using FLODANCE. The black curve (left axis, largely obscured) is the exact log-determinant $\ell_n$ computed by MEMDET. FLODANCE is fitted on $[1, n_s = 10^3]$ (yellow) and extrapolated to $[n_s, n = 10^4]$ (red). The blue curve (right axis) shows the relative error of prediction, which remains below 0.4%.

(non-homogeneous) linear model of coregionalization (LMC) (Gelfand et al., 2010, Section 28.7) given by

$$\kappa(\boldsymbol{x}, \boldsymbol{x}' \,|\, \alpha, \nu) = \sigma(\boldsymbol{x})^{\frac{1}{2}} \sigma(\boldsymbol{x}')^{\frac{1}{2}} \rho(\boldsymbol{x}, \boldsymbol{x}' \,|\, \alpha, \nu),$$

where $\sigma : \mathbb{R}^p \to \mathbb{R}^{d \times d}$ is the local covariance of the model's vector output of size $d$. In this model, we use the matrix square root, $\sigma^{\frac{1}{2}}$, to project the scalar Matérn correlation into the $d \times d$ coregionalization space while preserving positive-definiteness.

In our experiment, we generated $n = 10,000$ random spatial points in dimension $p = 2$ for a Gaussian process with output dimension $d = 10$, resulting in a covariance matrix of size $m = nd = 100,000$. We set the Matérn correlation scale to $\alpha = 0.04$ and the smoothness parameter to $\nu = 1.5$. The local covariance fields $\sigma(\boldsymbol{x})$ were instantiated as random symmetric positive-definite matrices drawn from a Wishart distribution.

Figure G.4 illustrates the effectiveness of FLODANCE on this Matérn-based covariance structure. Panel (a) shows the empirical scale law via the ratio of successive log-determinants, which exhibits a smooth trend consistent with the theoretical behavior observed for NTKs. Panel (b) evaluates the extrapolation accuracy of FLODANCE: the black curve denotes the

*Table H.1.* Computational complexity of log-determinant estimation methods. The first row corresponds to the exact method (MEMDET), while all other methods are approximations. One FLOP is counted as a fused multiply-add (FMA) operation.

| Method | Approach | Complexity | Description |
|---|---|---|---|
| MEMDET | Direct factorization | $\frac{1}{6}m^3 - \frac{1}{4}m^2 + \frac{1}{12}m$ | $m$ : Full matrix size, $m = nd$ |
| FLODANCE | Submatrix extrapolation | $\frac{1}{6}m_s^3 - \frac{1}{4}m_s^2 + \frac{1}{12}m_s + (q+3)^2 n_s$ | $n_s$ : Number of data samples from $n$ 
 $m_s$: Sampled matrix size, $m_s = n_s d$ 
 $q$ : Laurent series truncation order |
| SLQ | Stochastic trace estimation | $(m^2 l + m l^2)s$ | $l$ : Krylov subspace size 
 $s$ : Number of Monte Carlo samples |
| Pseudo NTK | Cross-class block reduction | $\frac{1}{6}\left(\frac{m}{d}\right)^3 - \frac{1}{4}\left(\frac{m}{d}\right)^2 + \frac{1}{12}\left(\frac{m}{d}\right) + m^2$ | $m$ : Full matrix size 
 $d$ : Number of model outputs |
| Block Diagonal | Class-wise block approx. | $\frac{1}{6}md^2 - \frac{1}{4}md + \frac{1}{12}m$ | $m$ : Full matrix size 
 $d$ : Number of model outputs |

*Table H.2.* Wall-clock runtimes (in seconds) for various log-determinant approximations $\hat{\ell}_n$, using the same models and configurations as in Table 2.

| | Model | Configuration | ResNet9 | ResNet9 | ResNet18 | MobileNet |
|---|---|---|---|---|---|---|
| **Quantity** | **Dataset** 
 **Subsample Size** | | CIFAR-10 
 $n = 1000$ | CIFAR-10 
 $n = 2500$ | CIFAR-10 
 $n = 1000$ | MNIST 
 $n = 2500$ |
| | Direct Computation (16-bit) | | 6.69 | 49.35 | 5.50 | 70.90 |
| | Direct Computation (32-bit) | | 7.08 | 49.39 | 5.87 | 54.05 |
| Runtime (seconds) | Direct Computation (64-bit) | | 7.09 | 51.57 | 5.97 | 51.22 |
| | SLQ | | 8.904 | 67.70 | 22.18 | 148.8 |
| | Block Diagonal | | 0.003 | 0.009 | 0.003 | 0.008 |
| | Pseudo NTK | | 0.015 | 0.077 | 0.014 | 0.082 |
| | FLODANCE | $n_0 = 1, \quad n_s = 50$ | 0.008 | 0.008 | 0.008 | 0.009 |
| | FLODANCE | $n_0 = 1, \quad n_s = 100$ | 0.017 | 0.018 | 0.016 | 0.017 |
| | FLODANCE | $n_0 = 300, n_s = 500$ | 0.377 | 0.350 | 0.354 | 0.376 |

exact log-determinant $\ell_n$ computed via MEMDET, while the colored curves show the fit (yellow) over the small interval $[1, n_s = 10^3]$ and the extrapolation (red) to $[n_s, n = 10^4]$. The right axis displays the relative error (blue), which remains below $0.4\%$ throughout. This example demonstrates FLODANCE's flexibility in handling more typical structured kernels.

## H. Comparison of Log-Determinant Methods: Complexity and Runtime

We summarize the computational characteristics of the log-determinant approximation methods evaluated in this work. Table H.1 compares their theoretical computational complexity, highlighting how each method accesses or approximates the full matrix—whether through exact computation with full matrix access (MEMDET), or through approximation strategies such as subsampling (FLODANCE), matrix–vector product oracles (SLQ), or blockwise approximations (Pseudo NTK and Block Diagonal).

Table H.2 reports wall-clock runtimes (in seconds) for several NTK datasets, using the same models and configurations as in Table 2. It compares direct 64-bit computations performed on matrices stored in 16-, 32-, and 64-bit precisions to various approximations, including FLODANCE, SLQ, and others. All measurements were conducted on the same hardware under comparable conditions. FLODANCE consistently achieves sub-second runtimes even on subsamples of size $n = 2500$, outperforming other approximations while avoiding the instability issues associated with SLQ.

# I. Implementation and Reproducibility Guide

We developed a Python package `detkit`[3] that implements the MEMDET algorithm and can be used to reproduce the numerical results of this paper. A minimalistic usage of the `detkit.memdet` function is shown in Listing I.1, where the user can specify various parameters: the maximum memory limit (`max_mem`), the structure of the matrix via the `assume` argument—set to `gen` for generic matrices (Algorithm D.1), `sym` for symmetric matrices (Algorithm D.2), and `spd` for symmetric positive-definite matrices (Algorithm D.3)—whether the data is provided in full or in its lower/upper triangular form (`triangle`), the arithmetic precision used during computation (`mixed_precision`), the location of scratchpad space on disk (`scratch_dir`), and enabling parallel data transfer between memory and disk (`parallel_io`). The function in this example returns the log and sign of the determinant of the full-size matrix (`ld`, `sign`), along with the diagonal entries of matrix $\mathbf{D}$ (`diag`) and the array of permutation indices (`perm`) for the permutation matrix $\mathbf{P}$ from the LDL decomposition $\mathbf{P}^\mathsf{T}\mathbf{M}\mathbf{P} = \mathbf{L}\mathbf{D}\mathbf{L}^\mathsf{T}$.

*Listing I.1.* A minimalistic usage of detkit package. The function memdet computes logabsdet($\mathbf{M}$) using the MEMDET algorithm.

```
    # Install detkit with "pip install detkit"
    from detkit import memdet
    import zarr

5   # NTK matrix M on disk
    M = zarr.open('filename.zarr', mode='r')

    # Compute logabsdet(M) and sgn(det(M)) with Algorithm D.2
    # Assume M is symmetric and only its upper triangle part is referenced.
10  ld, sign, diag, perm, info = memdet(M, max_mem='32GB', assume='sym', triangle='u',
                                        overwrite=False, mixed_precision='float64',
                                        scratch_dir='/tmp', parallel_io='tensorstore',
                                        verbose=True, return_info=True, flops=True)
```

The next example, shown in Listing I.2, demonstrates the use of the FLODANCE method via the `FitLogdet` class in `detkit`. This method fits a scaling law to a small subset of log-determinants computed from submatrices and extrapolates to larger submatrix sizes using Algorithm 1. Specifically, we use the `diag` array from Listing I.1 to compute the log-determinants $\ell_k = \sum_{i=1}^{kd} \log|D_{ii}|$, $k = n_0, \ldots, n_s$, for submatrices of size $kd \times kd$, where $d$ is the output dimension of the model (e.g., number of classes in CIFAR-10). These submatrices correspond to a permuted ordering of the original matrix during LDL decomposition, i.e., $\tilde{\mathbf{M}} := \mathbf{P}^\mathsf{T}\mathbf{M}\mathbf{P}$. While the sampling of submatrices could, in principle, be performed in any order, the LDL decomposition conveniently provides the log-determinants of successive principal submatrices—formed by selecting the first $kd$ rows and columns—at no additional cost; an advantage we exploit in this approach. The resulting sequence $\ell_k$ is then fit over the interval $k \in [n_0, n_s]$, and the fitted FLODANCE model is used to predict log-determinants in the extrapolation range $[n_s, n]$.

We have also concurrently developed a separate high-performance Python package, `imate`,[4] which implements stochastic Lanczos quadrature (SLQ), a randomized method for approximating the log-determinant at scale. This package is implemented with a C++/CUDA backend and supports execution on both CPU and multiple GPUs. Listing I.3 demonstrates a minimalistic usage of the `imate.logdet` function.

---

[3]`detkit` is available for installation from PyPI (https://pypi.org/project/detkit), the documentation can be found at https://ameli.github.io/detkit, and the source code is available at https://github.com/ameli/detkit.

[4]`imate` is available for installation from PyPI (https://pypi.org/project/imate), the documentation can be found at https://ameli.github.io/imate, and the source code is available at https://github.com/ameli/imate.

*Listing I.2.* The class `FitLogdet` fits and extrapolates log-determinants using FLODANCE in Algorithm 1.

```python
import numpy as np
from detkit import FitLogdet

# Range of datapoints (n) and number of labels (d) for CIFAR-10
n, d = np.range(50000), 10

# Compute ℓ_k := logabsdet(M̃_{[:k:k]}) for sub-matrices of the size k = 1,...,m = nd
# Here, diag is an array of length m obtained from Listing I.1
ell = np.cumsum(np.log(np.abs(diag)))

# Keep every d-th element
ell = ell[(d-1)::d]

# Choose a fit interval, such as (n_0, n_s) = (10^2, 5 × 10^3)
n0, ns = 1e2, 5e3
fit_mask = (n > n0) & (n < ns)

# Fit using Algorithm 1 with 4-th order truncated Laurent series
flodet = FitLogdet(q=4)
flodet.fit(n[fit_mask], ell[fit_mask])

# Extrapolate in a larger interval, such as in (n_s, n) = (5 × 10^3, 5 × 10^4)
n_eval = np.geomspace(ns, n)
ell_eval = flodet.eval(n_eval)
```

*Listing I.3.* A minimalistic usage of `imate` package. The function `logdet` computes $\mathrm{logdet}(\mathbf{M})$ using the stochastic Lanczos quadrature algorithm.

```python
# Install imate with "pip install imate"
from imate import logdet
import numpy

# Number of data (n) and labels (d)
n, d = 50000, 10

# NTK matrix M on disk
M = numpy.memmap('filename.npy', mode='r', dtype='float32', shape=(n*d, n*d))

# Compute logdet(M) using stochastic Lanczos quadrature (SLQ) method
# Assume M is symmetric.
ld, info = logdet(M, method='slq', min_num_samples=100, max_num_samples=200,
                  lanczos_degree=100, error_rtol=0.01, confidence_level=0.95,
                  outlier_significance_level=0.001, orthogonalize=-1, num_threads=0,
                  num_gpu_devices=0, gpu=True, verbose=True, return_info=True,
                  plot=True)
```

