# OpenReview forum: "Determinant Estimation under Memory Constraints and Neural Scaling Laws"
_ICML.cc/2025/Conference — ICML 2025 poster_

### Official Review · Reviewer_QjXZ · 2025-02-26

**Overall Recommendation:** 3

**Summary:**

This paper proposes a scalable way to compute Neural Tangent Kernel log-determinants of dense matrices which may arise in training deep neural networks. The empirical Neural Tangent Kernel has been shown as the effective tool to study the behavior of neural networks during both training and inference. In particular, the Gram matrix (i.e. the kernel matrix) can be used in lazy-training and in obtaining uncertainty quantification estimates. Computing log-determinants of such NTK kernels for a large $n$ is extremely hard and sensitive to small eigenvalues. The challenges include: 1) dealing with storing the large kernel matrix; 2) a scalable way to compute functions of the kernel matrix (in this case, the log determinant). The authors use an existing block $LU/LDL^T$ decompositions to efficiently load blocks of pre-computed Gram matrix from disk. However, realizing that this still poses a problem for a large value of $n$, the authors propose regressing over precomputed log determinants up to $p \times p$. For a large $n$, the authors propose an asymptotic approximation. The authors test the proposed approximation on the NTK matrices corresponding to widely used models such as ResNet9, ResNet18, and ResNet50.

## update after rebuttal
We thank the authors for their rebuttals. For the most part I am satisfied with the replies. I will maintain weak accept for the submission. In order for the submission to get a higher score (for this conference and for other follow-up work based on this submission), the authors are encouraged to:
- demonstrate some empirical comparison in computation time; the submission focuses on approximation quality mainly.
- tabulate symbols used in the writeup, as at least one other reviewer also mentioned difficulties in keeping track of the symbols.

**Claims And Evidence:**

The claims made in the paper seem to be sound and based on well-known linear algebraic techniques for computing decompositions which may be used to compute log determinants. A clever way to scale this up to $n$ by an asympotitic approximation is interesting.

**Essential References Not Discussed:**

I am not aware of other essential references which should have been included. However, it may be enlightening to include how the proposed method is different algorithmically from SLQ and Pseudo NTK approaches.

**Experimental Designs Or Analyses:**

Experimental designs are more or less sound, although for a non-expert, it is not too clear why the implied value of $m = 10$ is used for all of the experiments (I see this from the fact that $K_{1000}$ results in $10,000 \times 10,000$ matrix; see Section 4.1).

**Methods And Evaluation Criteria:**

The proposed benchmark datasets are well chosen and suitable for evalating the problem. The comparison against competing methods (such as SLQ and Pseudo NTK) are also included.

**Other Comments Or Suggestions:**

1. Table 1: SLQ  = Stochastic Lanczos Quadrature, but this is nowhere mentioned in the writeup until this point.
2. Section 3.2: shouldn't the precomputed sequence be: $\{ L_1, ... L_p \}$, not up to $L_n$?
3. In Algorithm 1: I was reading the index for all variables used in the regression equation for $y_n$. It was strange to see the index shifted by $+1$, but the formula for $y_n$ in Section 3.2 was shifted by $-1$. This needs to be consistent.

**Other Strengths And Weaknesses:**

Strengths:
1. Considers an important problem of interest in machine learning. Log determinant computation is inherently hard.
2. Scaling law consideration for approximation is interesting.

Weaknesses:
1. Some of the plots in the paper are not easy to understand (see below).
2. The difference between the proposed method and the competing methods such as SLQ should be highlighted. I could not find any information in the writeup itself.

**Questions For Authors:**

1. Why is $m = 10$ assumed in for all datasets in the experiment?
2. Are there any way you can use other variable names to help the readers, e.g. remembering the definition of $m$ across the entire writeup is challenging. The definition of $p$ is okay enough though.

**Relation To Broader Scientific Literature:**

The authors provide broader connection to other machine learning applications such as model selection, quantification of generalization bounds, etc. The considered problem of log-determinant computation and the proposed methodology is quite relevant to many of the interesting problems in machine learning.

**Theoretical Claims:**

I am familiar with the correctness of MEMDET algorithm since these are based on already-existing block decompositions of large matrices. I did check the appendix for proof of Proposition 1 and it seems to be correct.

---

> ### Author Rebuttal · Authors · 2025-04-01
>
> We thank the reviewer for their careful review of our paper and positive feedback on our work. We acknowledge that some points need clarification and aim to do so below.
>
> Regarding the implicit value of $m=10$: this is the number of classes used in all the classification datasets we used in experiments. We will state this explicitly in Section 4, as we understand the confusion.  We further thank the reviewer for their suggestion regarding plot and variable labeling, as well as algorithm naming. We have included these improvements and corrections in the updated version of the paper.
>
> Regarding Section D: this is included as algorithmic analysis of MEMDET. Many of these were conducted for our own benefit to choose appropriate block sizes, and to predict how long the computations would take. MEMDET is an exact algorithm (with no approximation occurring) and was required as a baseline for our experiments, since we could not fit the full NTKs into memory on the machines we were using. We believe that exact methods will always have a place, and so hope the information in Appendix D will be useful to others.
>
> To clarify the difference between algorithms: Pseudo-NTK summarizes the data by summing over rows and columns $m\times m$ block, then dividing by $m$.
> This gives a good approximation with respect to the operator norm.
> However, it should not be expected to give refined determinant estimates.
> Both SLQ and FLODANCE build their approximations from cheaper quantities.
> In the case of SLQ, matrix vector products are used in a Monte Carlo approximation.
> This gives a low bias estiamte with high variance, particularly when the condition number of the matrix is large.
> On the other hand, in FLODANCE we use extrapolation from matrix minors.
> This gives a low variance estimate in our experience (see http://anonymous.4open.science/r/memdet-E8C1/notebooks/scale_law_pred_uq.pdf), while the magnitude of the bias is unclear, and subject to the stationarity of the process in the training region (see http://anonymous.4open.science/r/memdet-E8C1/notebooks/stationarity.pdf).
> We agree that including more details of these methods will improve our work, and have added a short description in the main body, and a more in-depth section in the appendix of the updated version of the document.
>
> Please let us know if anything else needs verification or clarification, as we are eager to engage further if necessary to improve our work.

---

> > ### Comment · Reviewer_QjXZ · 2025-04-01
> >
> > Dear authors,
> >
> > Thank you for your reply. I have noted the clarifications that you have provided. For the most part I am satisfied with the replies. In order for me to raise the score further, this work needs to demonstrate:
> > - some empirical comparison in computation time; the submission focuses on approximation quality mainly.
> > - At least one other review has mentioned difficulties in keeping track of the symbols that are used. It would be helpful if these are tabulated somewhere in the writeup.
> >
> > At the moment - I will be maintaining the current score.

---

> > > ### Author Response · Authors · 2025-04-03
> > >
> > > We thank the reviewer for their quick response and further suggestions.
> > >
> > > First, regarding nomenclature: we are aware that in the submitted document there were some minor nomenclature clashes and inconsistencies. This has since been rectified, and a table containing our current conventions can be found at http://anonymous.4open.science/r/memdet-E8C1/notebooks/nomenclature.pdf.
> > >
> > > Regarding computation time, we refer the reviewer to the updated version of Table 1 which now includes compute time and cost, found at http://anonymous.4open.science/r/memdet-E8C1/notebooks/comparison.pdf.
> > > We have also made a companion table to Table 2 to report computation time at http://anonymous.4open.science/r/memdet-E8C1/notebooks/walltime_det.pdf.
> > >
> > > In order to further clarify the distinction between the different methods, we will also include the following table, outlining the computational complexity of the different methods we consider http://anonymous.4open.science/r/memdet-E8C1/notebooks/complexity.pdf.
> > > This shift from $m^3$ to $m_s^3$ complexity (for matrix and submatrix respectively of size $m\times m$ and $m_s\times m_s$) moving from LDL/MEMDET to FLODANCE means that when using say $10$% of the full dataset, roughly a $1000\times$ speedup is realized. While SLQ may seemingly offer an advantage here (when roughly $m_s^3>m^2sl$), this would require $ls<\frac{m}{1000}$ and we were unable to get good accuracy in this scenario.
> > > Instead we found that for NTK matrices the accuracy of SLQ was poor, even as these parameters grew such that the runtime was comparable to (and even exceeded) that of MEMDET.
> > >
> > > We remark that this scaling behavior and the corresponding speed improvement is only one benefit of our method.
> > > The fact that the full matrix does not need to be computed means that quantities that would otherwise be intractable can be accurately approximated.
> > > To see this, Table B.1 in http://anonymous.4open.science/r/memdet-E8C1/notebooks/memory.pdf shows the storage size required for the NTKs of some standard benchmark datasets used in machine learning.
> > > For context, estimates on the time required to form the NTK matrices themselves can be found in Table B.2 at http://anonymous.4open.science/r/memdet-E8C1/notebooks/memory.pdf.
> > > Since both storage and formation are quadratic in the number of datapoints, the ability to extrapolate from again $10$% of the data leads to $100\times$ faster NTK formation.
> > >
> > > Thank you for these suggestions, we are more than happy to include the subsequent improvements in the final version of our paper. We hope that this positively impacts your evaluation of our work.

---

### Official Review · Reviewer_ikyx · 2025-03-07

**Overall Recommendation:** 4

**Summary:**

The paper designs two algorithms for computing log-determinant of large PSD matrices under memory constraints. The first algorithm is named MEMDET, designed based on block LDL decomposition. The other one is named FLODANCE, designed based on neural scaling law assumptions. Empirical results show that the proposed methods achieve significant speedup while maintaining high accuracy.

**Claims And Evidence:**

The authors claim that the proposed methods achieves fast and accurate log-determinant estimation under memory constraints. The claims are well-supported through descriptions on algorithm design, complexity analysis and empirical studies.

**Essential References Not Discussed:**

The paper may benefit from additional discussions about the Lanczos family of matrix trace estimation techniques.

**Experimental Designs Or Analyses:**

The proposed methods are evaluated across various neural networks and sample sizes. They are practically sound but may benefit from more experiments beyond the NTK scenario.

**Methods And Evaluation Criteria:**

The proposed methods are evaluated under the NTK estimation scenario for deep learning. the evaluation criteria focus on accuracy, running speed and memory usage, aligning well with the studied problem.

**Other Comments Or Suggestions:**

No comments.

**Other Strengths And Weaknesses:**

Strength:
The proposed algorithm designs are promising, and the empirical performance gain of the proposed methods is significant.

Weakness:
The neural scaling law assumption may need further justification, e.g. empirical results on some ill-conditioned / noisy scenarios.

**Questions For Authors:**

* From Table 2, the estimation seems very unstable since the results vary a lot across different methods. How is the numerical stability of the proposed MEMDET method? If the ground truths in Table 2 are acquired by MEMDET, it seems not very fair since FLODANCE is using MEMDET as the backbone. How are these results compared to the native method? Also, I believe the absolute value of the log-determinant does not matter a lot, instead we are more interested in its relative value across different settings. From this point, showing the relative behavior of each method (e.g. verifying the neural scaling law) may give a more solid comparison.

**Relation To Broader Scientific Literature:**

The proposed methods can facilitate relevant researches on the practical or theoretical behavior of neural tangent kernels. They may also inspire new matrix-based algorithms for, e.g. kernel approximation.

**Theoretical Claims:**

I checked the proofs of the main theoretical results and they are rigorous. However, the proof is largely relying on the scaling-law assumption, which may require further justification.

---

> ### Author Rebuttal · Authors · 2025-04-01
>
> We thank the reviewer for their constructive feedback and positive review of our work. The reviewer has raised a few points that deserve addressing, and we believe that doing so will improve our work.
>
> The first of these is our dependence on the scaling law assumption. In the submitted version of the document, we provided criteria on the kernel which guarantees the scaling-laws hold in Appendix F. However, these are technical conditions, and their satisfaction is non-trivial to check (although we have discussed some possible verification in the response to Reviewer zrLg). This means that like most of the literature exploring scaling laws, we are relying on empirical behavior to check that the scaling laws are satisfied. To this end, http://anonymous.4open.science/r/memdet-E8C1/notebooks/stationarity.pdf contains a figure that compares the ratio of successive determinants to the fitted scaling law. Note that the log-scale on the $y$-axis turns this into a residual plot. These residuals are approximately normally distributed, which is evidence that Assumpsions 1 and 2 are satisfied. This plot has been included in the updated version of our document, along with discussion.
>
> Regarding the focus of our numerics on the NTK: this was simply done because these matrices are known to be a particularly difficult class of kernel matrices to work with. For modern neural networks, the corresponding kernels are non-stationary, and the matrices tend to
> be highly ill-conditioned and very large, often not being feasible to compute explicitly: hence the
> need for our algorithm. We believe that this makes NTK matrices particularly useful examples
> to highlight the utility and efficacy of our method.
> However, we do recognize that other Gram matrices are commonly used in machine learning applications. In particular, we computed the Gram matrix associated to the linear model of coregionalization [1], with 10 outputs and a Matern kernel over 10,000 data points. Plots demonstrating the performance of FLODANCE on this dataset, with experimental details, can be found at http://anonymous.4open.science/r/memdet-E8C1/notebooks/matern.pdf .
> We have included this experiment in the appendix of an updated version of our work.
>
> Other reviewers have also drawn attention to the lack of the distinction between our methods and their competitors. To this end, we have included a brief section in the  the appendix outlining the stochastic Lanczos methods, as well as the Pseudo-NTK. We believe this will help our exposition, in addition to a brief higher-level summary in the main document.
>
> Thank you for your suggestions regarding Table 2. An updated version of this table containing relative error can be found at http://anonymous.4open.science/r/memdet-E8C1/notebooks/relative.pdf . We remark that MEMDET is an exact method that allows for determinant computation of large matrices that do not fit into memory. It achieves this through block-decompositions, relying on an underlying algorithm (LDL$^\top$, LU, or Cholesky) that is amenable to this type of decomposition. As such, the use of MEMDET in our FLODANCE approximations reduces to LDL$^\top$, since the training sample is small enough to fit in memory. We have emphasised this in the final version of the document.
>
> Please let us know if anything else needs verification or clarification, as we are eager to engage further if necessary to improve our work.
>
> [1] https://doi.org/10.1007/BF02066732

---

> > ### Comment · Reviewer_ikyx · 2025-04-01
> >
> > Thanks for the response and the additional experiments. I understand that MEMDET is an exact method, but what I am curious about is its numerical stability. Even for exact methods, the accumulating rounding error in each step can still lead to a large bias in the result, especially for ill-conditioned problems like NTK. Given the current results, it seems that there are already non-negligible differences between 32-bit and 64-bit results, indicating that the method may be numerically unstable. Have you compared your method with some native ones like direct eigenvalue decomposition, perhaps for some small $n$ so it could fit into memory? How do their results differ between 32-bit and 64-bit computations?

---

> > > ### Author Response · Authors · 2025-04-06
> > >
> > > We thank the reviewer for their engagement and for raising this question. Before addressing it, we would like to clarify what we mean by *precision* in our paper, as it may help disentangle the origin of numerical differences across settings.
> > >
> > > Our computational pipeline consists of three stages:
> > >
> > > 1. **Training stage:** We trained all neural networks (e.g., ResNet50, ResNet9) in 32-bit precision, which is the default and standard practice in most deep learning frameworks like PyTorch.
> > >
> > > 2. **NTK computation stage:** The NTK matrix is computed from the trained model and stored in various precisions (e.g., 16-bit, 32-bit, and 64-bit, from the same pre-trained model). The "precision" of the NTK matrix, as referred to throughout our paper, reflects the compute and storage format at this stage. Due to the large expense of forming these matrices, it is often tempting or necessary to form and store these matrices in lower precisions. Our low-precision experiments highlight the pitfalls of mixed-precision in these cases as per Section 2.1, regardless of the downstream use case.
> > >
> > > 3. **Log-determinant computation stage:** Regardless of how the NTK matrix was computed and stored (16-bit, 32-bit, or 64-bit), *all* log-determinant computations were performed in 64-bit precision, across all methods: MEMDET, SLQ, FLODANCE, etc. This presents the ``best-case'' mixed-precision scenario.
> > >
> > > Since MEMDET entirely eliminates memory requirement barriers, it became practical for us to perform high-precision computations (e.g., 64-bit in stage 3) even on large matrices—thus mitigating common concerns about overhead associated with higher-precision formats.
> > >
> > > **Clarifying Differences Between 32-bit and 64-bit Results.**
> > > Log-determinants computed from NTK matrices generated in 32-bit and 64-bit precision can differ, but these differences arise from the input matrices themselves being different (i.e., generated under different floating-point formats upstream, as discussed in Section 2.1). For a fixed input matrix, all exact log-determinant methods—whether based on MEMDET or eigenvalue decomposition—produce nearly identical results.
> > >
> > > To illustrate this, we performed two evaluations:
> > >
> > > 1. In this comparison (https://anonymous.4open.science/r/memdet-E8C1/notebooks/eig_small.pdf), we compared MEMDET against both NumPy’s `slogdet` and `eigh` on the same matrices. The results match closely, with relative errors comparable to the differences between `slogdet` and `eigh` themselves.
> > >
> > > 2. In this figure (https://anonymous.4open.science/r/memdet-E8C1/notebooks/comp_eig_ldl.pdf), we extended this comparison to NTK matrices of increasing size (up to $\sim 82,000$), in both 32-bit and 64-bit forms. For a fixed matrix, all three methods produce log-determinants consistent up to relative errors in the range of $10^{-12}$ to $10^{-7}$. Differences between the 32-bit and 64-bit plots reflect differences in the NTK matrix itself, not instability in MEMDET or any of the other methods.
> > >
> > > This confirms that any variation in log-determinants across precision settings is attributable to upstream matrix formation, not the methods used for computing log-determinants.
> > >
> > > **On Actual Stability Challenges and Our Solution.**
> > > That said, we do want to highlight one real stability issue we encountered and addressed. Although NTK matrices are theoretically positive definite, when computed in lower precision (especially 16-bit or 32-bit), small eigenvalues may flip sign near zero, causing Cholesky decomposition to fail. MEMDET avoids this issue by using LDL decomposition instead—a numerically stable alternative that shares the same computational cost but is more robust in the presence of ill-conditioning (see also the end of Section 2.2 on p.4 of our submitted document).
> > >
> > > Furthermore, we validated that MEMDET’s block-wise LDL decomposition matches the standard (in-memory) LDL decomposition to machine precision. This is illustrated in this figure (https://anonymous.4open.science/r/memdet-E8C1/notebooks/vary_num_blocks.pdf), where we varied the number of memory blocks used by MEMDET. The results show relative errors consistently around $10^{-14}$, reinforcing that our method maintains high numerical fidelity regardless of memory constraints.
> > >
> > > As part of our ongoing effort to improve the completeness of experiments, the larger-scale NTK matrices used in Table 1 and Section 4.2 are being updated. Specifically, the NTK matrix for ResNet9 (at full size $500,000$) has now been recomputed in 64-bit precision, and the NTK matrix for ResNet50 will be updated shortly using a full-size matrix of $500,000$ in 64-bit (currently being processed). As a reminder, all log-determinant computations on the large matrices in these experiments were already performed in full 64-bit precision using MEMDET.
> > >
> > > We hope this clarifies the distinction between differences due to matrix precision and the stability of MEMDET itself, and has had a positive impact on your assessment of our work.

---

### Official Review · Reviewer_zrLg · 2025-03-11

**Overall Recommendation:** 5

**Summary:**

This work addresses the memory and computation bottlenecks in estimating log-determinants of large matrices such as the Neural Tangent Kernel (NTK) for large models and datasets. The paper introduces MEMDET, a memory-constrained algorithm for exact log-determinant calculations for matrices too large to fit in memory by using block LU decompositions with an efficient block ordering to minimize data transfer. The paper then proposes FLODANCE, an accurate algorithm for extrapolating log-determinants computed on a small subset of the dataset to the much larger full dataset by exploiting the scaling behavior of the log-determinants for a class of kernels.

**Claims And Evidence:**

Yes

**Essential References Not Discussed:**

None that I'm aware of.

**Experimental Designs Or Analyses:**

All experiments in Section 4 appear sound.

**Methods And Evaluation Criteria:**

Yes

**Other Comments Or Suggestions:**

While FLODANCE performed strongly in the experiments relative to alternatives, it's unclear how useful the absolute level of accuracy it achieves is for practical applications (e.g. model selection). Can the authors compare FLODANCE with alternatives in some downstream applications that require estimating the log-determinants?

**Other Strengths And Weaknesses:**

Strength: Overall, the paper is very well presented. The conceptual insights and experimental results are both strong.

Weakness: Three nontrivial assumptions are required to prove the scaling law (Theorem F.1), but they were not verified in the experiments. In this case, it's unclear how the theory presented in Section 3.1 can actually explain the observed scaling behavior of the log-determinants.

**Questions For Authors:**

1. Can you motivate the form of the non-asymptotic correction to the exponent $\nu$? How much does it affect the accuracy of the estimate?

**Relation To Broader Scientific Literature:**

In contrast to conventional numerical methods for estimating log-determinants such as SLQ, I find FLODANCE's approach of exploiting the scaling behavior of the log-determinants of a specific class of kernels for efficient estimation quite creative and novel. Analyzing the scaling behaviors under various limits has shown to be a powerful tool in machine learning broadly, such as predicting optimal hyperparameters at scale [1]. This paper demonstrates that this idea can inspire similar algorithmic progress in numerical linear algebra.

[1] Tensor programs v: Tuning large neural networks via zero-shot hyperparameter transfer. Yang et al. 2022

**Theoretical Claims:**

I checked the proofs in Appendix F and G.

---

> ### Author Rebuttal · Authors · 2025-04-01
>
> We thank the reviewer for their constructive feedback and positive review of our work. This work is indeed part of a broader active research program aimed at providing computational tools for estimating linear algebraic quantities at scale. We will make this more clear in the final version of our paper, and cite the related work  for hyperparameter selection that you suggested.
>
> Regarding the assumptions underlying the scaling law behavior, we acknowledge that these technical assumptions have been identified but not verified.
> To do so for the empirical neural tangent kernel matrix may not be realistic, however, we have come across [1], which examines the spectral decomposition of the analytic NTK (an approximation of the empirical variant we consider) on the sphere. In [1, Theorem 4.8], it appears that our Assumption F.1 is verified over a bounded eigenbasis (Assumption F.2), and Assumption F.3 is automatically satisfied as we consider $f_p^\ast = 0$. If desired, we are happy to include this discussion in the final manuscript.
> More generally, however, the theory is instead used to establish plausibility of the scaling law, since there is a well-characterized class of kernels for which these assumptions hold. We then use the scaling laws as an empirical theory in much the same way that others do in the ML literature.
>
> In terms of this empirical behavior, a plot of the residuals of the successive $\log$-determinant increments after removal of the scaling law can be found at http://anonymous.4open.science/r/memdet-E8C1/notebooks/stationarity.pdf.
> This figure demonstrates that these values are approximately normally distributed, empirically validating the scaling law in Assumption 1 (as well as Assumption 2).
> This figure is included in the updated version of our work, and clarification has been added regarding the distinction between this empirical verification and the assumptions stated in Appendix F.
>
> Regarding downstream applications, we note that the $\log$-determinant is typically only one of multiple terms appearing in quantities of interest. For this reason, we believe it is premature to include studies computing these partial quantities as approximations, until we have developed the toolkit to treat the other terms.
>
> The non-asymptotic terms that appear in the exponent model sublinear behavior that occurs in the pre-limit.
> To be clear, they model the function $\nu = \nu(n)$ by allowing for subsequent terms beyond the constant in its Laurent series.
> These terms were included for flexibility as we found that they do improve accuracy for matrices at the scale of our experiments.
> However, they are not strictly necessary: when the exponent is treated as a constant in $n$, we found that our method still outperforms SLQ (see http://anonymous.4open.science/r/memdet-E8C1/notebooks/relative.pdf for a version of Table 2, updated to display relative error for reviewer ikyx).
> Errors and cost also now appear in an updated version of Table 1 found at http://anonymous.4open.science/r/memdet-E8C1/notebooks/comparison.pdf . Algorithm 1 has also been updated to explicitly state how the terms are computed.
>
> Please let us know if anything else needs verification or clarification, as we are eager to engage further if necessary to improve our work.
>
> [1] Murray, M., Jin, H., Bowman, B., \& Montufar, G. (2023, February). Characterizing the Spectrum of the NTK via a Power Series Expansion. In International Conference on Learning Representations.

---

### Official Review · Reviewer_EHZs · 2025-03-14

**Overall Recommendation:** 3

**Summary:**

This work proposes a method to scale the calculation of matrix determinant to extremely large matrices, especially in context of ill-conditioned matrices such as the empirical NTK matrix. They first develop a memory-constrained algorithm which computes determinant exactly and can serve as a baseline for their later experiments. The core idea behind their main method is to derive an scaling law for the determinant and then use the obtained fit to extrapolate to determinants of even larger matrices. The approach is quite interesting and, besides helping with the particular problem, might also inspire techniques that utilize extrapolation for computation of spectral quantities pertaining to neural networks.

**Claims And Evidence:**

- The experimentation is all specific to NTK. But the method is pitched as a method which is generally applicable. While that's true, I don't think there is strong empirical evidence to support this claim. Especially the scaling law approach may not work in other contexts.

Also, see methods and evaluation criteria section

**Essential References Not Discussed:**

--

**Experimental Designs Or Analyses:**

In general, the experimental design seems fine but I have a few questions about them:

- what is the floating point precision used for the baselines such as block diagonal, SLQ, Pseudo NTK?

- do all the methods under comparison use the same sized subset?

- what is p in the table 2? also, i suppose memdet baseline is the L_n one? if so, please label it more explicitly, you don't remember all possible symbols in your first few reads :)

**Methods And Evaluation Criteria:**

- It would have been more interesting to see the use of determinant in some specific context, and then observe that a better estimation leads to better improvements. But, you know, often it's not the value of determinant in isolation that is needed, and rather some other quantities too. And then, in deep learning, things can always turn out a bit weird, so would have been nice to ground it somehow.

- Another thing I am not fully sure is if you just want to calculate determinant, how many samples of the dataset must be used to do so. This is very unclear. It could be that all estimates using 1000 samples are terrible, but you really need like 50K samples to get to the most accurate estimates. Therefore the gold standard is a bit hard to gauge. Perhaps the authors can do some testing in an appropriately downscaled setting, where say the determinant is exactly computed, in 64 bit precision, on all datapoints in the training set. Right now it is unclear how much to read in their FLODANCE estimates closer to the memdet baseline.

- Finally, it is also unclear how specific are the determinant estimates to the 'seed'-subset used for the scaling laws. See the questions section below for elaboration.

**Other Comments Or Suggestions:**

It's not clear to me how much the details of MEMDET are relevant to a ML audience. Maybe put some of that in appendix, and describe the scaling law and its theory a bit better.

**Other Strengths And Weaknesses:**

Overall the paper is well written, and the idea of using scaling laws is clever. But how accurate and robust these determinant estimates are and how much the downstream applications gains from it are far from clear.

**Questions For Authors:**

- How specific are the determinant estimates to the subset used for deriving the scaling laws? Can you present some experimental evidence which would support that your estimate is robust to this change? On a similar note, how big must the subset be to allow for this robustness?

 - I believe the scaling laws has been shown in the context of LLMs trained in an online manner. Is there evidence that similar scaling laws also hold in the classical vision settings as used here? Otherwise that should be stated as an assumption as well.

**Relation To Broader Scientific Literature:**

I think the use of scaling laws could be an interesting direction for numerical estimation, which I believe is relatively unexplored.

**Theoretical Claims:**

--

---

> ### Author Rebuttal · Authors · 2025-04-01
>
> We would like to thank the reviewer for taking the time to review our paper, and for their constructive feedback.
> Regarding general applicability of our methods: we stress that MEMDET computes the exact (64-bit) determinant, and is a general method that works for arbitrary matrices.
> On the other hand, our neural scaling law approach, FLODANCE, is only suggested to work for Gram matrices satisfying Assumption 1.
> In our writeup and experiments, we focus on NTK matrices, since they are known to be a particularly difficult class of kernel matrices. Other algorithms (e.g. SLQ) already perform very well on better behaved, more traditional classes of matrices.
> For modern neural networks, the corresponding NTK kernels are non-stationary, and the Gram matrices tend to be highly ill-conditioned and very large, often not being feasible to compute explicitly: hence the need for our algorithm.
> We believe that this makes NTK matrices particularly useful examples to highlight the utility and efficacy of our method.
> However, we do recognize that other Gram matrices are commonly used in ML applications. In particular, we computed the Gram matrix associated to the linear model of coregionalization [1], with 10 outputs, and Matern kernel over 10,000 data points. Plots demonstrating the performance of FLODANCE on this dataset, along with experimental details, can be found at http://anonymous.4open.science/r/memdet-E8C1/notebooks/matern.pdf .
>
> As the reviewer has pointed out, the $\log$-determinant often appears as one term in more elaborate quantities of interest.
> Examples of this are the quadratic-form term in the $\log$-marginal likelihood of a Gaussian process (along with their gradients for the purposes of training), and the curvature term that appears in the IIC.
> Estimating these quantities is a clear goal.
> However, the other terms appearing in these quantities are highly challenging to estimate as well, requiring separate novel techniques that we believe will each be of independent interest.
> This work is part of a broader research program under active development, and the current submission already contains two novel techniques for computing and estimating a crucial term, with corresponding software.
> We believe this strikes an appropriate balance of utility, impact and digestibility for an outlet such as ICML.
>
> To clarify the sensitivity of FLODANCE to subsample size and seed subset, we have conducted the following experiments.
> First, http://anonymous.4open.science/r/memdet-E8C1/notebooks/param_selection.pdf shows the effect of the number of training samples on prediction accuracy, and describes the experimental setup.
> As is to be expected, the general trend is that using more samples results in better prediction.
> There is a natural tradeoff between computation time and accuracy that is up to the practitioner to determine.
>
> Second, we tested the sensitivity of the estimate to the training sample. To this end, we sampled an ensemble of 15 submatrices of the NTK for ResNet-50 trained on CIFAR10. Each submatrix contians 10,000 datapoints with 10 classes, forming a $100,000\times100,000$ NTK matrix. Experimental details and plots can be found at http://anonymous.4open.science/r/memdet-E8C1/notebooks/scale_law_pred_uq.pdf
>
> Third, we upgraded our computations on both architectures to use the full CIFAR-10 dataset with $(n = 50,000)$, resulting in NTK matrices of size $(m = 500,000)$. For ResNet9, we also recomputed log-determinants using 64-bit precision via MEMDET. The updated results are provided in http://anonymous.4open.science/r/memdet-E8C1/notebooks/scale_law_fit.pdf (was Figure H.1) and http://anonymous.4open.science/r/memdet-E8C1/notebooks/scale_law_pred.pdf (was Figure 4). The 64-bit computation for ResNet50 is underway and will be completed in a few days, with updated results included in the final version.
>
> We note that while the new plots are derived from the same NTK kernel and methodology, their appearance differs from earlier versions due to a technical adjustment. To improve quantization in 32-bit precision, we had previously scaled the NTK matrices, giving an additive term proportional to $n$ in the $\log$-determinant. Hence, the curves differed from the current ones by a linear function of $n$. This shift does not affect the fitting behavior or predictive quality, but it alters the shape of the curve. In the updated 64-bit computations, this scaling was no longer necessary, resulting in cleaner and more interpretable visualizations without any transformation applied.
>
> As is evident by these additional studies, FLODANCE is quite robust to these factors.
> We thank the reviewer for suggesting these experiments, as we believe their inclusion in the final version of the document will improve the quality of our work. Please let us know if anything else needs verification or clarification, as we are eager to engage further if necessary to improve our work.
>
> [1] https://doi.org/10.1007/BF02066732

---

### Decision · Program_Chairs · 2025-05-01

**Decision:**

Accept (poster)

**Comment:**

This paper proposes a method to calculate the determinants of large matrices which arise as Neural Tangent Kernel for large neural networks. The reviewers agree that the method is sound and most praised the originality and creativity of the work. After the discussion period, all reviewers now give positive scores. Although reviewer QjXZ initially complained of the lack of run time comparison to some other methods, the authors have provided a very convincing rebuttal with additional experiments, which should be included in the camera ready version. Reviewer EHZs also complains of the application being limited to NTK type matrices by the application of the scaling laws, this is an issue that can be solved by better writing and contextualization in the revision.
Since two reviewers complained of the difficulty following the notation, I recommend adding a table of notation.

Overall, this paper makes a very solid contribution and should be **accepted**.